# Overexpression of *Saussurea involucrata* dehydrin gene *SiDHN* promotes cold and drought tolerance in transgenic tomato plants

**Xinyong Guo, Li Zhang, Xiaozhen Wang, Minhuan Zhang, Yuxin Xi, Aiying Wang, Jianbo Zhu**⊙*

Key Laboratory of Agricultural Biotechnology, College of Life Science, Shihezi University, Shihezi, China

\* jianboz9@sina.com

**Data Availability Statement:** All relevant data are within the manuscript.

**Funding:** This work was supported by the National Science Foundation Project grant no. 31360053

## Abstract

Dehydrins are late embryogenesis abundant proteins that help regulate abiotic stress responses in plants. Overexpression of the *Saussurea involucrata* dehydrin gene *SiDHN* has previously been shown to improve water-use efficiency and enhance cold and drought tolerance of transgenic tobacco. To understand the mechanism by which *SiDHN* exerts its protective function, we transformed the *SiDHN* gene into tomato plants (*Solanum lycopersicum* L.) and assessed their response to abiotic stress. We observed that in response to stresses, the *SiDHN* transgenic tomato plants had increased contents of chlorophyll a and b, carotenoid and relative water content compared with wild-type plants. They also had higher maximal photochemical efficiency of photosystem II and accumulated more proline and soluble sugar. Compared to those wild-type plants, malondialdehyde content and relative electron leakage in transgenic plants were not significantly increased, and $H_2O_2$ and $O_2^-$ contents in transgenic tomato plants were significantly decreased. We further observed that the production of stress-related antioxidant enzymes, including superoxide dismutase, ascorbate peroxidase, peroxidase, and catalase, as well as pyrroline-5-carboxylate synthetase and lipid transfer protein 1, were up-regulated in the transgenic plants under cold and drought stress. Based on these observations, we conclude that overexpression of *SiDHN* gene can promote cold and drought tolerance of transgenic tomato plants by inhibiting cell membrane damage, protecting chloroplasts, and enhancing the reactive oxygen species scavenging capacity. The finding can be beneficial for the application of *SiDHN* gene in improving crop tolerance to abiotic stress and oxidative damage.

## Introduction

Harsh environmental conditions, including extreme temperatures, saline-alkaline conditions, and drought, can affect plant growth, development, and productivity [1]. More insidiously, these abiotic stresses can also result in excessive generation of reactive oxygen species (ROS), triggering oxidative-induced damages, such as oxidation of protein, DNA, or lipid, destruction of photosynthetic pigments, and inactivation of photosynthetic enzymes, thereby further

and the National transgenic major projects grant no. 2016ZX080005004-009 to JZ. The funders had no role in study design, data collection and analysis, decision to publish, or preparation of the manuscript.

**Competing interests:** The authors have declared that no competing interests exist.

threatening cellular viability [2–4]. Plants have developed powerful, multifaceted physiological and biochemical mechanisms that concertedly help sense, detoxify, eliminate, and/or neutralize ROS overproduction. Currently, there is a great deal of interest in identifying the genes conferring the greatest degrees of stress tolerance as targets for genetic engineering of crop plants.

Late embryogenesis abundant (LEA) proteins are closely linked with the environmental stress tolerance of plants and, as their name suggests, are accumulated during the late stages of seed development [5–6]. Dehydrins (DHNs) are the most commonly described group of hydrophilin LEA proteins. They have high hydrophilicity, containing a high proportion of charged and polar amino acids, but a low fraction of hydrophobic, non-polar residues; they also often lack cysteine and tryptophan [7–8]. Based on the number and distribution of conserved sequences (i.e., Y-, S-, and K-segments), dehydrins are subdivided into five types: $K_n$, $SK_n$, $Y_nSK_n$, $Y_nK_n$, and $K_nS$ [9–10].

Dehydrins are known to promote plant stress tolerance [9]. Earlier studies have reported that dehydrins were active in ion sequestration and membrane stabilization, and as chaperones [11–13]. More recent studies have identified, in a series of papers, that dehydrins also have ROS scavenging abilities [14–21]. For instance, Halder et al [16–18] showed that overexpression of *Sorghum bicolor* dehydrin genes in transgenic tobacco promoted temperature and osmotic stress tolerance by scavenging superoxide anions ($O_2^-$) and conferring protective effects to the enzymes responsible for dismutation of $O_2^-$. Similarly, Liu et al [20] reported that the expression of *ZmDHN13*, a maize dehydrin gene induced by hydrogen peroxide ($H_2O_2$), led to decreases in $O_2^-$ generation in transgenic tobacco plants subjected to oxidative stress. Cao et al [21] also demonstrated that overexpression of *Hevea brasiliense* dehydrins, *HbDHNs*, in *Arabidopsis thaliana* led to increased tolerance to salt, drought, and osmotic stresses of the plant. They also found that the plants exhibited higher activity of antioxidant enzymes and lowered accumulation of ROS. In our previous study, we also identified that dehydrin (*SiDHN*) from *Saussurea involucrata* Kar. et Kir. (a perennial herb that grows in the cold, high alpine mountains of central Asia) facilitated abiotic stress tolerance [22]. Despite these findings, the precise mechanism of action of dehydrins in promoting plant stress tolerance remains obscure [9]. Additionally, the effects of *SiDHN* gene on cold and drought tolerance of transgenic tomato plants have not been determined. Therefore, in this work, we overexpressed *S. involucrata* dehydrin *SiDHN* gene in transgenic tomato (*Solanum lycopersicum* L.) plants, one of the most important horticultural crops across the globe. We then determined the plants' tolerance to cold and drought stress by evaluating agronomic traits, changes of chlorophyll content, morphology and physiology of the plants. We also determined gene expression and activity of antioxidant enzymes involving in plants' responses to cold and drought stress, as well as the accumulation of $H_2O_2$ and $O_2^-$.

## Materials and methods

### Plant materials and growth conditions

Plantlets of *S. involucrata* were prepared by tissue culture from sterile *S. involucrata* leaf (explant) cut into small pieces with sizes of about 1.5–2 cm. To produce callus, the explant was grown on callus induction medium MSB supplemented with 6-BA (6-benzyladenine; 0.05 mg/L) and 2,4-D (2,4-dichlorophenoxyacetic acid; 0.3 mg/L), and the medium was changed every 15 d. After that, they were transferred to callus differentiation medium MS supplemented with 6-BA (2.0 mg/L) and NAA (α-naphthaleneacetic acid; 0.5 mg/L), during which the medium was changed every 15 d. Small buds obtained after callus differentiation were cut and transferred to rapid propagation medium MS containing 6-BA (0.5 mg/L) and NAA (0.01 mg/L) to

obtain plantlets. The tissue culture was carried out in a laboratory incubator on a 21˚C/19˚C day/night schedule, with a 16 h light/8 h dark cycle and at a light intensity of 70 µmol m$^{-2}$ s$^{-1}$.

Seeds of tomato plant variety 'Yaxin 87-5' (wild-type) were provided by Yaxin Seed Co. Ltd. (Shihezi City, Xinjiang, China). The wild-type tomato plants were grown from the seeds in our laboratory and were used to produce transgenic tomato plants. The plants were initially grown in a tissue culture room at 25˚C with a 12 h light/12 h dark cycle, 60–70% humidity, and a light intensity of 70 µmol m$^{-2}$ s$^{-1}$. After that, the plantlets were transplanted in plastic pots containing composite substrates consisting of equal parts of peat, vermiculite, and soil and then were allowed to grow in a naturally lit greenhouse at 22–28˚C and relative humidity of 60–70%.

## Plant treatments

Cold treatment of *S. involucrata* plantlets was carried out based on Liu et al [23], by which six-week-old *S. involucrata* plantlets were kept at 4˚C for 48 h, followed by 0˚C for additional 48 h. The drought treatment was carried out according to Zhu et al [24]. In this treatment, after plants were washed thoroughly with tap water to eliminate substrates and their surface water was wiped off, they were dehydrated on filter papers for 48 h. Untreated *S. involucrata* plantlets were used as a control. All plants were cultured in parallel in an incubator with 21˚C day/19˚C night cycle. Following the treatments, plant leaves at the same position (on each plant) were harvested at 0, 1, 3, 6, 9, 12, 24, and 48 h.

Cold treatment of tomato plants is similar to that of *S. involucrata* plantlets. Three wild-type and nine $T_2$ transgenic tomato plants (all were ten weeks old) were kept at 4˚C for 48 h, followed by 0˚C for additional 48 h. For drought treatment, the plants were grown without water for 24 days and thereafter were re-watered for recovery. After the treatments, the plants were grown at 25˚C in a greenhouse and were watered at about 1 L per basin once a week. Tomato leaves (the second and third leaves from the top) were collected after 48 h of the cold treatments (4$^o$C and 0$^o$C), or after 0, 16 and 24 d of the drought treatment. Immediately after harvest, the leaves were frozen in liquid nitrogen and stored at -80˚C until subsequent experiments.

All experiments were repeated three times, and each sample was prepared in triplicate.

## RNA extraction and quantitative RT–PCR analysis

Total RNA was isolated using the RNAisoPlus kit (TaKaRa) with on-column DNase I treatment, according to the manufacturer's instructions. First-strand cDNA was synthesized from the total RNA using oligo(dT) primers and PrimeScript®RTase (TaKaRa) enzyme. The expression patterns of *SiDHN* gene and ROS-related stress-responsive genes were subsequently determined by quantitative RT-PCR, which was carried out using a LightCycler 480 instrument (Roche Diagnostics, Basel, Switzerland) and SYBR® Premix Ex Taq™ (TaKaRa, China), according to the supplier's instructions and the MIQE guidelines [25]. Each qRT-PCR reaction mixture (10 µL) was composed of 50 ng of cDNA, 5 µL of 2 × SYBR® Green Master-Mix Reagent (Applied Biosystems) and 0.2 µM of the corresponding gene-specific primers shown in Table 1. For the analysis of *SiDHN* gene, the qRT-PCR thermal cycles were set as follows: 95˚C for 30 s, followed by 40 cycles of 95˚C for 5 s, and 60˚C for 25 s. The expression level of the gene was normalized to that of the *GAPDH* gene from *S. involucrate* (Accession No.: KF563904.1). For the analysis of ROS-related stress-responsive genes, the qRT-PCR thermal cycles were set as follows: predenaturation at 94˚C for 4 min, followed by 94˚C for 40 s, 30 cycles of 95˚C for 5 s, and 52˚C for 40 s. The expression levels of these genes were normalized to those of an internal control, the *SlEF1α* gene (GenBank ID: X53043) [26]. For the analysis of

**Table 1. Sequences of primers used for qRT-PCR analysis.**

| Primer | GenBank ID | Sense (5'→3') | Antisense (5'→3') |
|--------|-----------|---------------|-------------------|
| *SiDHN* | KJ145794 | TGTGGTGGAGCAGATCAAGG | TATGTCCATCGGCGACACCATGA |
| *SikGAPDH* | KF563904.1 | TAGCAAGGATGCTCCCATGTT | GGAGCAAGGCAGTTGGTTGTG |
| *SlEF1α* | X53043 | GGAACTTGAGAAGGAGCCTAAG | CAACACCAACAGCAACAGTCT |
| *SlAPX* | NM_001247853 | GATGTTCCCTTTCACCCTG | CCCCTCTTTTTCCCCACT |
| *SlCAT* | NM_001247898 | GGTGGATTATTTGCCCTCG | ACCTCTCCCCTGCCTGTTT |
| *SlPOX* | NM_001247203 | CACATACATTTGGAAGGGC | TTTATTGTTGGATCAGGGC |
| *SlSOD* | NM_001247840 | GGCCAATCTTTGACCCTTTA | AGTCCAGGAGCAAGTCCAGT |
| *SlP5CS* | U60267.1 | ACGGTCTTTACAGTGGTC | AAGCAGCATACATAGCAG |
| *SlLTP1* | NM_001247806.2 | TCTAGGAGGCTGTTGTGGTG | GTGGAGGGGCTGATCTTGTA |

Primers were designed at the 5′- and 3′- ends of the full-length DNA sequences retrieved using the corresponding GenBank IDs.

*SlEF1α* gene, the qRT-PCR thermal cycles were as follows: denaturation at 95˚C for 30 s, followed by 50 cycles at 95˚C for 5 s, 60˚C for 10 s, and 68˚C for 10 s. Each analysis was repeated three times, and three replicate reactions were prepared for each analysis. The relative gene expression was calculated using the ΔΔCt method previously described [27].

## Isolation and sequence analysis of Full-Length *SiDHN* cDNA

The construction of full-length *SiDHN* cDNA library was conducted using the Creator TMSMARTTM cDNA Library Construction Kit (Clontech, USA) as described previously [28]. Ninety-six single cells containing *SiDHN* were randomly selected and then subjected to plasmid extraction. After that, the plasmid was sequenced, and the identity of the inserted cDNA was confirmed by DNA sequencing [The obtained cDNA sequence showed homology to a gene encoding a water stress-induced dehydrin-like protein known as *SiDHN* gene]. The 5′ and 3′ ends and the CDS region of the *SiDHN* gene were identified using ORF finder by which the sequence was aligned with known sequences in the databases. Protein secondary structure and intrinsic disorder of the deduced SiDHN protein were predicted using PHYRE2 Protein Fold Recognition Server [29]. The obtained SiDHN amino acid sequence was searched against the NCBI database using Standard Protein BLAST. Dehydrin protein sequences from a total of 7 species including three most similar species (*Cynara cardunculus* var. *scolymus* (XP_024979388.1), *Artemisia annua* (PWA89867.1) and *Lactuca sativa* (XP_023764814.1), denoted as CsDHN, AaDHN and LsDHN, respectively) and four species of Solanum (*Solanum habrochaites* (AHB20199.1), *Solanum chilense* (ADQ73964.1), *Solanum peruvianum* (ADQ73992.1) and *Solanum lycopersicum* (NP_001316365.1), denoted as ScDHN, SpDHN, SlDHN and ShDHN, respectively) were retrieved. The sequences were then subjected to multiple sequence alignments using DNA-MAN8.0 software [30], and the identification of Y-, K-, and S-segments of the sequence was carried out using the PROSITE database [31]. A phylogenetic tree was constructed by the Neighbor-Joining method with 1,000 bootstrap replicates using MEGA 5.1 software (http://www.megasoftware.net/); bootstrap scores <50% were deleted.

## Plant transformation and transgenic tomato identification

In construction of pBIN438-SiDHN plasmid, *SiDHN* gene fragment was amplified from cDNA using the forward primer 5′- GGATCCAAAATGGCACAACACGGATA -3′ (the underline sequence indicates the *Bam*HI site) and the reverse primer 5′-GTCGACATGAAATGATT ACTTCTGACCCC -3′ (the underline sequence indicates the *Sal*I site). The amplicon were ligated with plant expression vector pBIN438 at the *Bam*HI and *Sal*I sites, and the obtained

plasmid was transformed into *Escherichia coli* DH5 alpha. The plasmid was transformed into *Agrobacterium tumefaciens* GV3101 using the method described previously [32]. A single cell was selected from the medium containing kanamycin, gentamicin and rifampicin, and the identity of the inserted gene was confirmed by DNA sequencing.

To produce transgenic tomato plants, the wild-type tomato plants were transformed with *SiDHN*-containing plasmid by Agrobacterium-mediated method. Briefly, leaves from the middle part of eight- to ten-day-old tomato plants were punched out to obtain leaf discs of about 0.5×0.5 cm$^2$ and used as the explant. The leaf discs were cultured on MS medium supplemented with ZT (zeatin; 1.0 mg/L) and IAA (indole-3-acetic acid; 0.3 mg/L) in the dark for 3 d. Subsequently, the cultured leaves were co-cultured with activated pBIN438-SiDHN bacterial liquid for 15 min; after that, they were removed and placed on a sterile filter paper to remove the bacterial liquid. The infected leaves were cultured for 3 d on a culture plate containing solid MS medium supplemented with ZT (1.0 mg/L) and IAA (0.3 mg/L) covered with a sterile filter paper. After 3 days, the leaves were transferred to a bud differentiation medium MS containing ZT (1.0 mg/L), IAA (0.3 mg/L), kanamycin (80 mg/L) and Tim (timentin; 100 mg/L), followed by 1 successive generation of about 15 d. A single bud was cut off and then cultured in a rooting medium (1/2 MS) containing IAA (0.6 mg/L), Kanamycin (80 mg/L) and Tim (100 mg/L) for 1 month. After that, the plants were transferred to a pot and allowed to grow in the conditions described in the 'Plant materials and growth conditions' section.

Primary (T0) tomato transformants were verified by PCR and T1 transgenic lines were selected by qRT-PCR both using the forward primer 5′−GGATCCAAAATGGCACAACACGGATA−3′ (the underlined section indicates the *BamHI* site) and the reverse primer 5′−GTCGACATGAAATGATTACTTCTGACCCC−3′ (the underlined section indicates the *SalI* site), synthesized by the Beijing Genomics Institute (TIANGEN Biotech Co. Ltd., Beijing, China). The T$_2$ homozygous lines were selected by segregation analysis as described by Pino et al [33], and three independent T$_2$ generation homozygous lines (denoted as OE-2, OE-5, and OE-8) with comparatively high expression levels of *SiDHN* were selected for further studies.

## Measurement of chlorophyll pigment content

Chlorophyll is responsible for absorption, transfer and conversion of light into energy in photosynthesis. When plants are subjected to stress, the content and composition of chlorophyll pigment are altered, which affects its photosynthetic ability [34]. In the measurement of chlorophyll pigment content, 0.1 g of leaves from WT and transgenic tomato treated with cold and drought conditions were cut into strips of 1-mm width and then added to test tubes. After that, 15 ml of acetone:ethanol mixture (V:V = 1:1) was added to extract chlorophyll. The extraction was performed at room temperature in the dark until the leaf strips became white. After centrifugation, the supernatant was subjected to absorbance measurements at 470 nm, 645 nm, and 663 nm, and the contents of chlorophyll a (Chl a), chlorophyll b (Chl b), and carotenoids (Car) were calculated by the following equations: Chl a = 12.21$^*$A$_{663}$ − 2.81$^*$A$_{645}$; Chl b = 20.13$^*$A$_{645}$ − 5.03$^*$A$_{663}$; and Car = (1000$^*$A$_{470}$ − 3.27$^*$Chl a − 104$^*$Chl b)/229.

## Determination of morphological and physiological traits

Various morphological and physiological traits, including leaf relative water content (RWC), malondialdehyde (MDA) content, relative electrolyte leakage (REL), maximal photochemical efficiency of photosystem II (PSII), and contents of proline and total soluble sugar (soluble osmotic regulators), were measured at different stages after both stress treatments.

Leaf relative water content is a physiological index used to evaluate the water retention capacity and is a parameter that can appropriately measure water status and osmotic

adjustment of plants under abiotic stresses [35]. RWC was determined following the method described by Lara et al [36] and was calculated by the following equation: RWC = (FW − DW)/(TW − DW) × 100%, where FW is the fresh weight of the leaves, TW is the weight at full turgor (measured after floating the leaves in distilled water for 24 h under light at room temperature), and DW is the weight after drying the leaves at 70˚C until a constant weight was achieved.

Malondialdehyde is a product of lipid peroxidation and is an important indicator determining the destructive effects of ROS and membrane damage upon stress [37]. Thus, measurement of MDA content can reflect the degree of membrane damage. Leaves were excised from the tomato plants and then washed with deionized water. After that, leaf discs were punched out, and the MDA contents in the leaf discs were quantified by the modified thiobarbituric acid reaction outlined in Du et al using a spectrophotometer (UV-160A, Shimadzu Scientific Instruments, Japan) [38].

In a plant system, abiotic stresses primarily target cell membrane, interfering with the membrane stability and integrity, and relative electrolyte leakage (REL) is a parameter generally used to indicate such interference [39]. Relative electrolyte leakage was determined following the method of Du et al using an EC 215 Conductivity Meter (Markson Science Inc., Del Mar, CA, USA) [38]. Young, fully-grown leaves were randomly selected and then subjected to electrolyte leakage analysis using a conductivity meter. The relative conductivity was calculated using the following equation: REL = (C1 − CW)/(C2 − CW) × 100, where C1 is the electrical conductivity value during the first measurement, C2 the conductivity value after boiling, and CW the conductivity of deionized water.

Maximal photochemical efficiency of photosystem II (PSII) is the efficiency at which light is absorbed by PSII. It determines the photochemistry of PSII when all of its reaction centers are open. Maximal photochemical efficiency of PSII ($F_v/F_m$) is a parameter indicating the degree of photoinhibition of PSII and its structural integrity [40]. The $F_v/F_m$ value of the tomato leaves was measured using a portable fluorescence analyzer (DUAL-PAM-100, Walz, Germany). Leaves were allowed to adapt to darkness for 30 min and then exposed to a flash of saturated light for 1 s. The minimal fluorescence ($F_0$) in the dark-adapted state (or when all PSII reaction centers are open) and the maximal fluorescence ($F_m$) during exposure to the saturated light pulse (or when all PSII reaction centers are closed) were measured, and the variable fluorescence ($F_v = F_m − F_0$) was calculated [41].

Proline and soluble sugars are soluble osmotic regulators that regulate and maintain integrity in plant cell membrane; thus, their contents can be used to indicate membrane integrity [42–43]. The content of free proline were measured according to the method described by Bates et al. [44]. Briefly, about 200 mg of ground leaf was extracted with 4 mL of 3% sulphosalicylic acid at 100˚C for 10 min. The homogenate was centrifuged at 12,000×g for 2 min, and 2 mL of the supernatant was then mixed with 2 mL of acid-ninhydrin reagent and 2 mL of glacial acetic acid. The mixture was boiled for 30 min and then soaked in an ice bath to terminate the reaction. The mixture was extracted with toluene (4 mL), and an absorbance at 520 nm of the organic phase was determined. To determine proline content, the absorbance values were compared with a standard curve constructed using known amounts of proline. Measurement of total soluble sugar content was carried out using the anthrone method in which glucose was used as the standard [45]. Approximately 200 mg of ground leaf was homogenized in 1 mL of distilled water and then boiled for 20 min. After centrifugation at 13,000×g for 10 min, 2 mL of the supernatant was mixed with 1.8 mL of distilled water and 2.0 mL of 0.14% (w/v) anthrone solution in 100% $H_2SO_4$ and then soaked in boiling water for 20 min. After cooling, an absorbance at 620 nm of the mixture was determined. The absorbance values were compared with a standard curve constructed using known amounts of glucose.

Each experiment was carried out in triplicate, and three replicate samples were prepared in parallel.

## Determination of antioxidant enzyme activity and content of reactive oxygen species

After exposure to cold and drought treatments, 0.5 g of fresh $T_2$ wild-type and transgenic tomato plant leaves were collected. The leaves were cut into small pieces and then homogenized in 4 mL of 50 mM sodium phosphate buffer (pH 7.8) containing 1% polyvinylpyrrolidone and 10 mM β-mercaptoethanol in an ice bath. The homogenate was centrifuged at 17,426×g for 15 min at 4˚C. The enzyme activity of the supernatant was then determined. Ascorbate peroxidase (APX) activity was determined following the method of Nakano and Asada [46] as the decrease in absorbance at 290 nm of ascorbate. Catalase (CAT) activity was determined according to the method described by Cakmak et al [47]. Superoxide dismutase (SOD) activity was assessed by an absorbance at 560 nm based on the methods of Beauchamp and Fridovich [48]. Peroxidase (POX) activity was determined according to the method described by Doerge et al [49]. All absorbance measurements were carried out on an Infinite M200 Pro microplate reader (Tecan Group Ltd., Männedorf, Switzerland).

Contents of $H_2O_2$ and $O_2^-$ were determined using a standard curve following the method described by Benikhlef et al [50], in which an absorbance at 415 and 530 nm was measured using a UV-160A spectrophotometer (Shimadzu Scientific Instruments, Japan).

## Statistical analysis

Statistical analysis of the data was performed using GraphPad Prism 7.0 and SigmaPlot 12 (SYSTAT Software). Data are expressed as means ± standard deviations of three replicate experiments. The expression level at 0 h (control time point) was defined as 1.0. Significant differences between the wild-type and transgenic plant lines were determined using Dunnett's multiple comparisons test, and $P < 0.05$ and $P < 0.01$ are considered significantly different.

## Results

### Sequence analysis of *SiDHN* gene from *S. involucrata*

The sequence analysis showed that the full-length cDNA of *SiDHN* consists of 703 base pairs with a 333-base-pair open reading frame, encoding a protein of 111 amino acids (GenBank accession No. KJ145794). It contains one Y-segment "TDEYVHNP" and two K-segments ("HEQGGKGV VEQI" and "HEKKGVMEKIKEKIPG"; Fig 1A); the latter are found in nearly all dehydrins [51]. In addition, *in silico* secondary structure and intrinsic disorder prediction using PHYRE2 showed that SiDHN protein comprises 36% α-helices located mainly in the Y-segment and both of the two K-segments, and over 78% of its amino acid residues are disordered.

Protein sequence alignment revealed that the deduced protein sequence of SiDHN shared 46%, 38%, and 42% similarity with dehydrin proteins from *C. cardunculus var. scolymus* (XP_024979388.1), *A. annua* (PWA89867.1), and *L. sativa* (XP_023764813.1), respectively (Fig 1B). By contrast, SiDHN protein is rather different from dehydrin proteins from four species of Solanum [23]: it shared only 19.4% similarity with *S. habrochaites* (AHB20199.1), 15.6% similarity with *S. chilense* (ADQ73964.1), 15.6% similarity with *S. peruvianum* (ADQ73992.1), and 15.0% similarity with *S. lycopersicum* (NP_001316365.1). The phylogenetic tree (Fig 1C) further showed that these proteins belong to three ($YK_2$, $YSK_2$, and $SK_3$) of the five types of dehydrins: SiDHN protein belongs to $YK_2$ type and is closely related to dehydrins from *C. cardunculus* var. *scolymus*, *Lactuca sativa* and *Artemisia annua* genes, which belong to $YSK_2$ type.

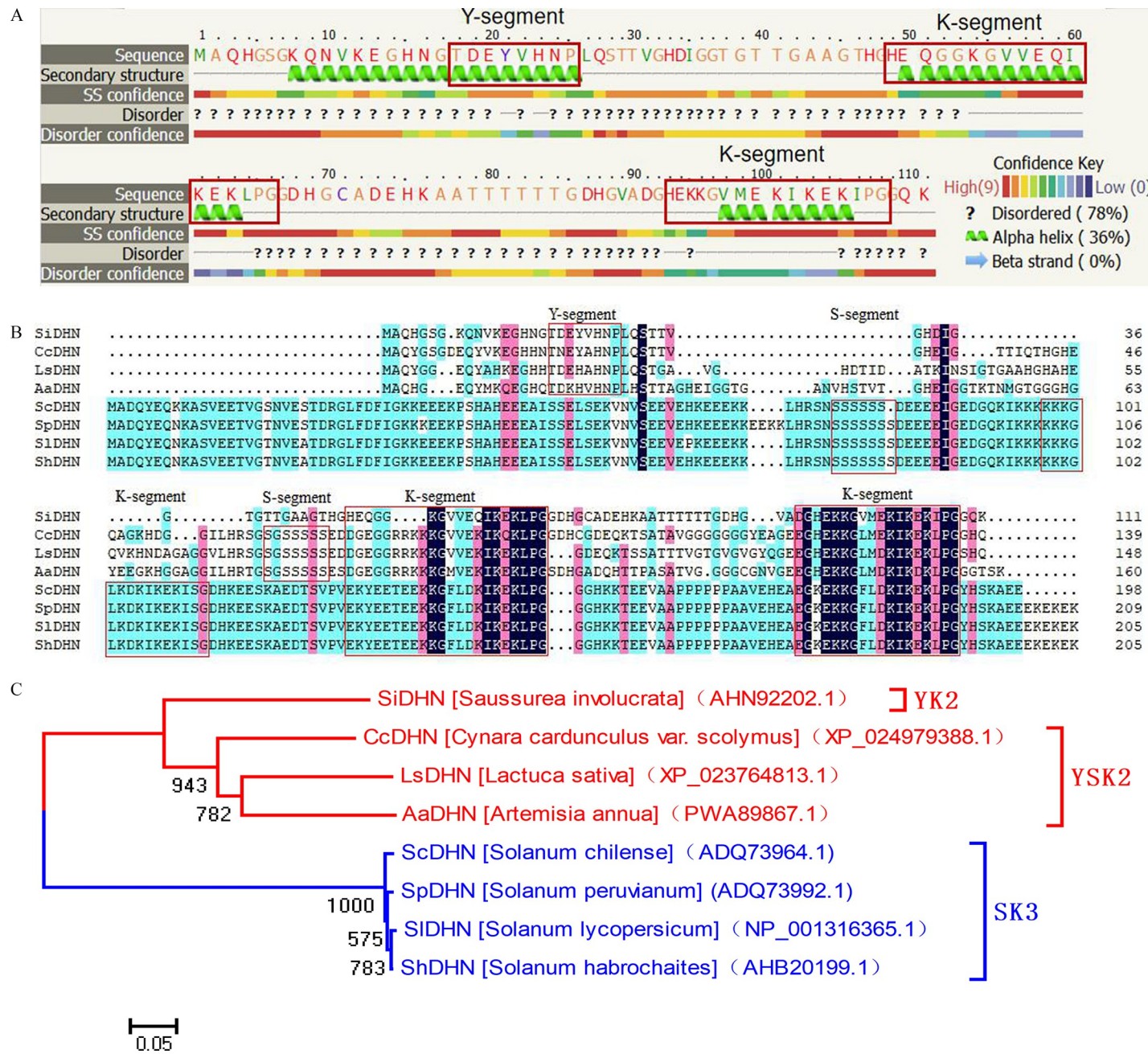

**Fig 1. Sequence analysis of *SiDHN* gene from *Saussurea involucrata*.** (A) Protein secondary structure and intrinsic disorder of deduced SiDHN protein predicted by PHYRE2 protein fold recognition server. (B) Protein sequence alignment of SiDHN protein with dehydrin proteins from *Cynara cardunculus* var. *scolymus* DHN (XP_024979388.1), *Lactuca sativa* DHN (XP_023764814.1), *Artemisia annua* DHN(PWA89867.1), *Solanum habrochaites* DHN (AHB20199.1), *Solanum chilense* DHN (ADQ73964.1), *Solanum peruvianum* DHN (ADQ73992.1) and *Solanum lycopersicum* DHN (NP_001316365.1). (C) Phylogenetic tree of SiDHN and dehydrins from species of Solanum and other Asterid members analyzed by the Neighbor-Joining method in MEGA 5.1. Bootstrap value was obtained from 1000 replicates.

## Expression of *SiDHN* gene in *S. involucrata* under cold and drought stress

At both temperatures, the gene expression levels of *SiDHN* in leaves of *S. involucrata* under cold stress steadily increased with increasing treatment time, reaching maximum values at 48 h (Fig 2A and 2B), and decreased thereafter (Data not shown). The expression in plants treated

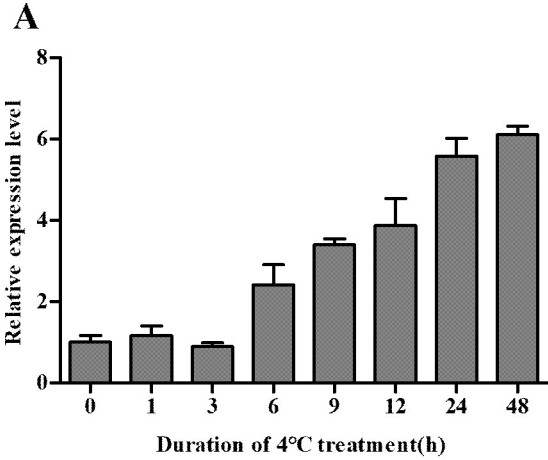
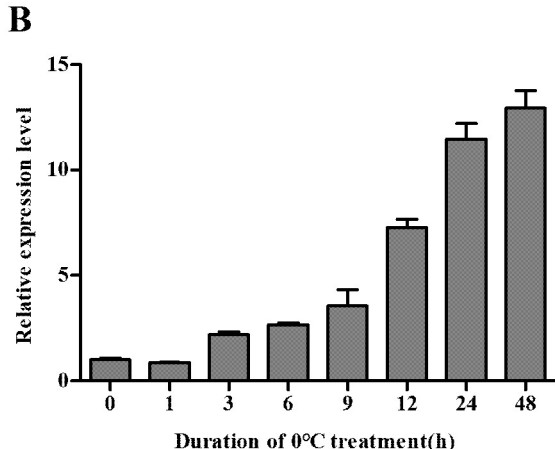
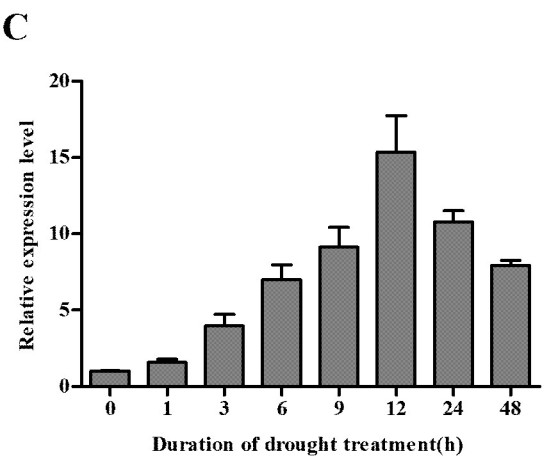
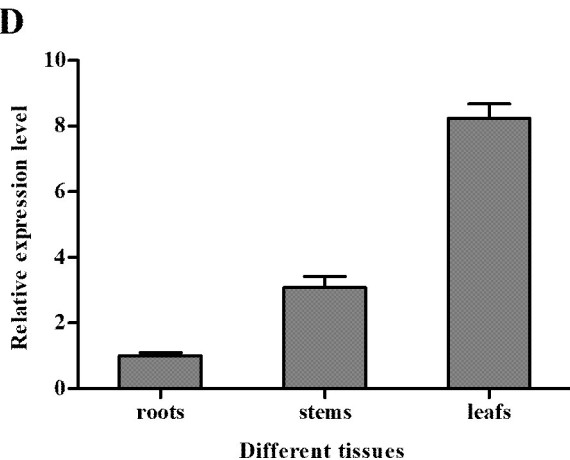

**Fig 2. Relative expression of *SiDHN* gene in *Saussurea involucrata* under cold and drought stress.** (A) Expression under cold treatment at 4˚C for 0, 1, 3, 6, 9, 12, 24, and 48 h. (B) Expression under cold treatment at 0˚C for 0, 1, 3, 6, 9, 12, 24 and 48 h. (C) Expression under drought stress for 0, 1, 3, 6, 9, 12, 24 and 48 h. (D) Expression in different plant tissues under cold stress. The values are means from 3 repeated experiments.

with 0˚C for 48 h were 10 times higher than those at 0 h and were nearly double those in the plants treated with 4˚C for 48 h. Similarly, the expression of *SiDHN* gene in *S. involucrata* treated with drought was also increased with increasing treatment time, reaching a peak value only after 12 h (Fig 2C), while gradually declined thereafter. Nonetheless, the expression at 48 h remained high with a value of 7 times higher than that at 0 h. The *SiDHN* was also found to constitutively express in roots and stems of plant under cold stress; however, the level in leaves was significantly higher than that in other two tissues (Fig 2D).

## Transformation and molecular characterization of *SiDHN*-overexpressing transgenic tomato plants

Following the method described in Pino et al [33], 30 independent kanamycin-resistant primary transformant plants (T$_0$ generation) were generated, and the successful transformation of the *SiDHN* gene was confirmed by PCR amplification. Among these transformant plants, 15 were confirmed by PCR amplification using a CaMV35S forward and gene-specific reverse

primer pair, and 10 were found to posses kanamycin resistance and have a 3:1 segregation ratio in the $T_1$ generation. Three independent homozygous lines (OE-2, OE-5, and OE-8) that had higher expression levels of *SiDHN* were selected for further studies (Fig 3).

### Agronomic traits of *SiDHN*-overexpressing transgenic tomato plants

Comparison of 80-day-old wild-type and *SiDHN*-overexpressing transgenic tomato plants showed that the leaves of the transgenic plants had denser and darker color than the wild-type plants (Fig 4), and each of the transgenic plants was shorter than the wild-type plants (Fig 5A). The fresh weights of both types of plants were not different; however, dry weights of the transgenic plant lines were higher than those of the wild-type plants (Fig 5B and 5C). Length of root (Fig 5D) and fresh and dry weight of root (Fig 5E and 5F) of the transgenic plants were greater than those of the wild-type plants. The transgenic tomato plants also had higher stem diameters than the wild-type plants (Fig 5G).

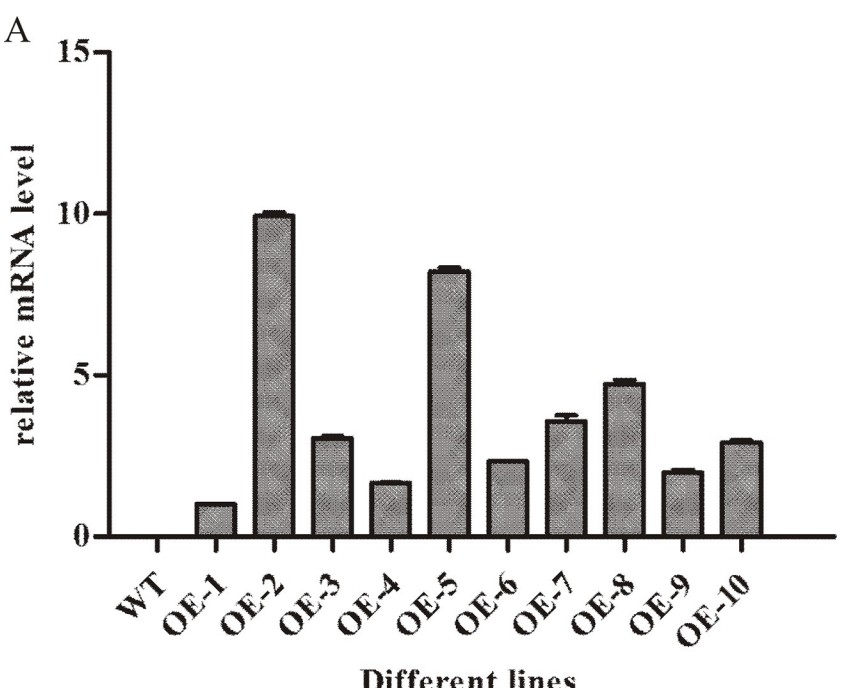

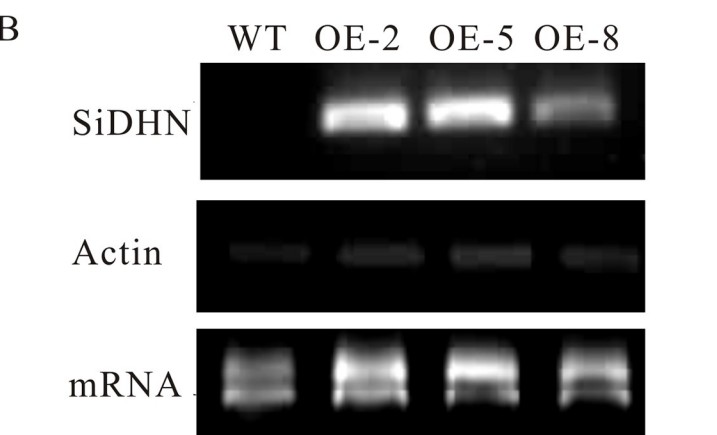

**Fig 3. Expression levels of *SiDHN* in wild-type and *SiDHN*-overexpressing transgenic tomato plant lines.**

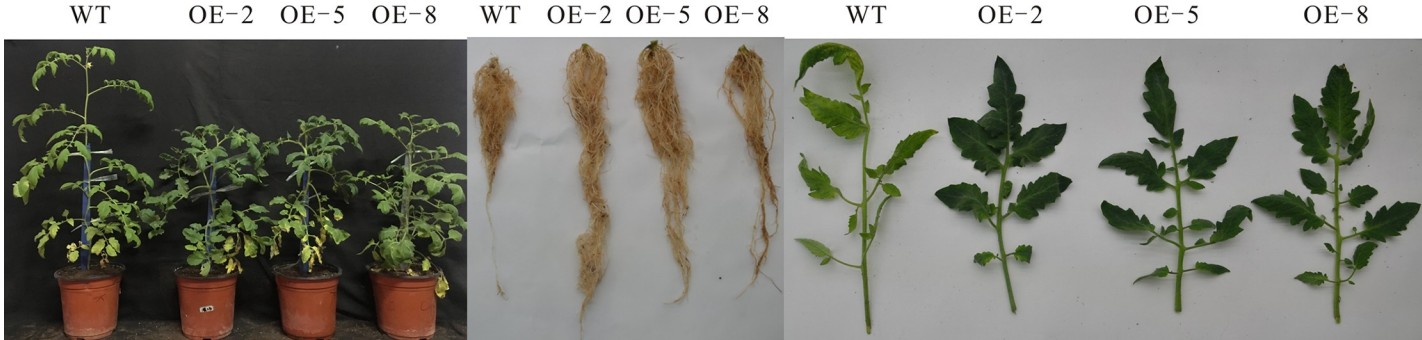

**Fig 4.** Phenotypes of the aboveground parts (Left), underground parts (Middle), and leaves (Right) of wild-type and *SiDHN*-overexpressing transgenic tomato plant lines (OE-2, OE-5, and OE-8).

## Chlorophyll pigment content in *SiDHN*-overexpressing transgenic tomato plants

Chlorophyll a (Chl a) levels in the transgenic plants were significantly higher than those in the wild-type plants (Fig 6A). Chlorophyll b (Chl b) content in the OE-2 and OE-8 lines was significantly greater that that in the wild-type plants. The Chlb content in OE-5 line was also greater, but the difference was not significant (Fig 6B). The Chl a+Chl b content in all three transgenic plant lines was higher than that in the wild-type plants (Fig 6C). The carotenoid (Car) content in the three transgenic plants lines was also higher than the wild-type plants, but the differences were only significant between the OE-2 and OE-8 plant lines and the wild-type plants (Fig 6D).

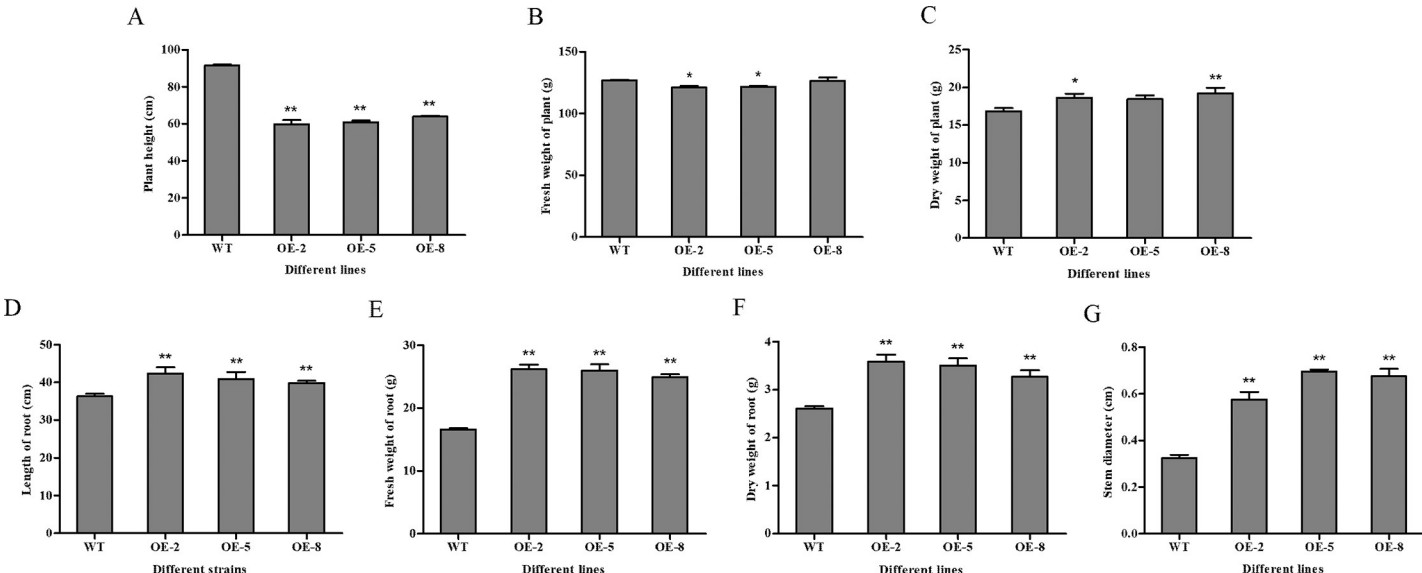

**Fig 5. Agronomic traits of 80-day-old wild-type and *SiDHN*-overexpressing transgenic tomato plant lines (OE-2, OE-5, and OE-8).** (A) Plant height. (B) Fresh weight of plant. (C) Dry weight of plant. (D) Length of root. (E) Fresh weight of root. (F) Dry weight of root. (G) Stem diameter. Data are means ± SD of three replicate experiments. Asterisk(s) indicate significant difference between the wild-type and transgenic plants: * represents $P < 0.05$ and ** represents $P < 0.01$.

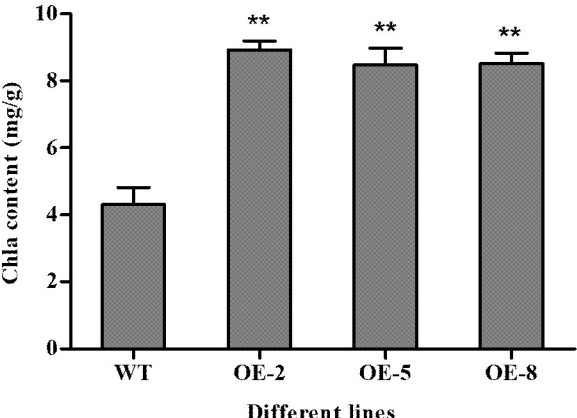
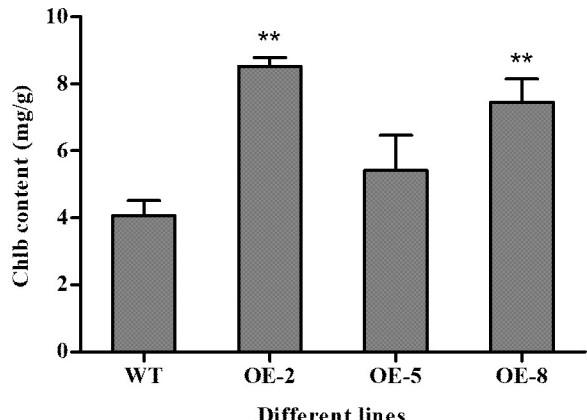
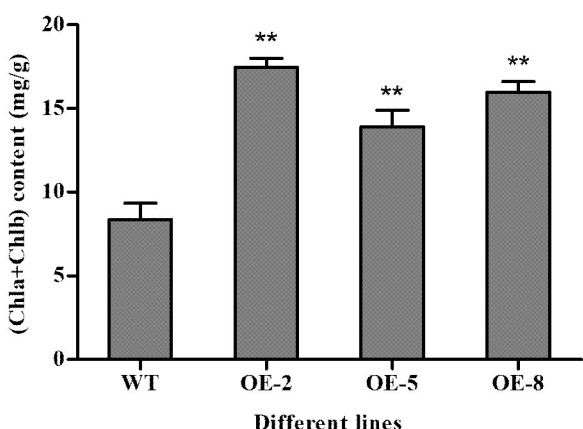
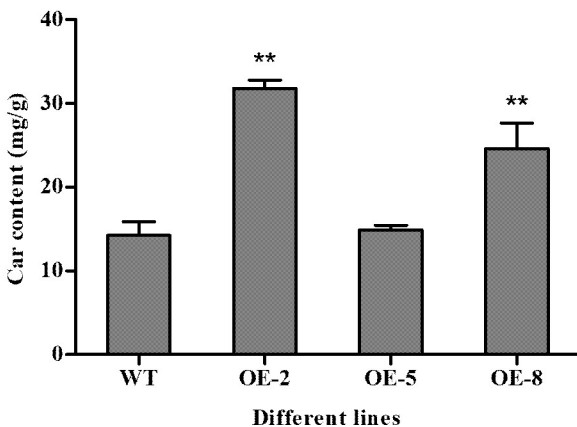

**Fig 6. Chlorophyll pigment content in wild-type and *SiDHN*-overexpressing transgenic tomato plant lines (OE-2, OE-5, and OE-8).** (A) Chl a content. (B) Chl b content. (C) Chl a+Chl b content. (D) Car content. Data are means ± SD of three replicates. Asterisk(s) indicate significant difference between the wild-type and transgenic plants: * represents $P < 0.05$ and ** represents $P < 0.01$.

## Morphological changes of *SiDHN*-overexpressing transgenic tomato plants under cold and drought stress

The phenotypes of the transgenic and wild-type plants at room temperature (25˚C) were different. After being exposed to 4˚C for 48 h, the wild-type plants exhibited signs of severe wilting; in contrast, the transgenic plants showed only slight signs of leaf wilting. After exposure to 0˚C for 12 h, the leaves of the wild-type tomato turned a deeper green color and signs of wilting became more severe, whereas the leaves of the transgenic plants exhibited no substantial changes (Fig 7A).

Aside from the difference in height, which was insignificant, no significant phenotypic differences were observed between the transgenic and wild-type plants under drought stress (Fig 7B). After 16 d without water, however, the wild-type plants exhibited clear signs of wilting, with the leaves near the base particularly flaccid and yellowed; by contrast, the leaves of the transgenic tomato plants showed minimal wilting with only a few yellowish leaves. This pattern was maintained after 24 d, but the OE-8 plants were also considerably more wilted and shorter than other transgenic plants. After 3 d of re-watered, the wild-type plants had recovered, but the stems were soft and limp. The OE-8 plants were also recovered, while the OE-2

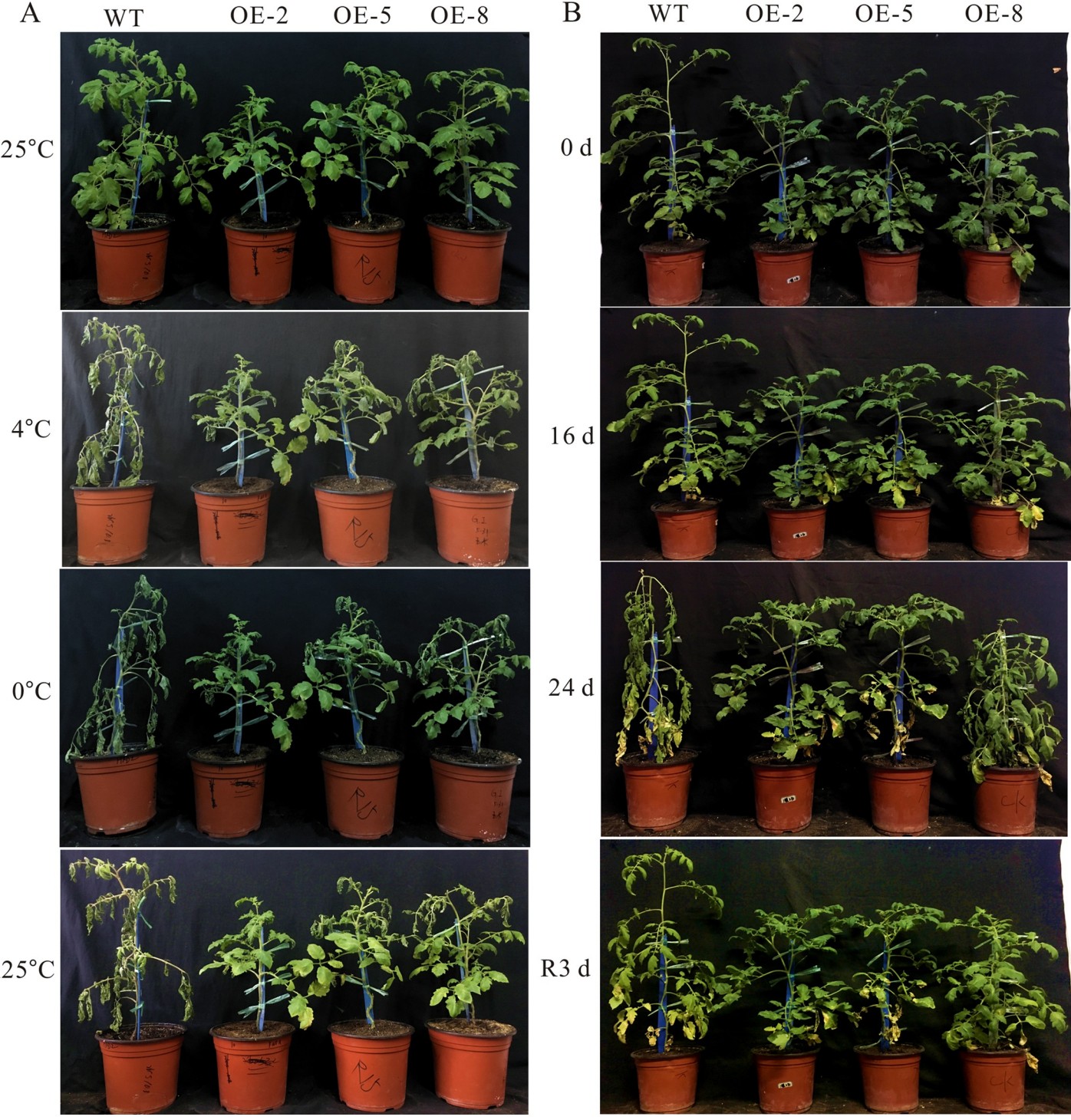

**Fig 7. Phenotypes of wild-type and *SiDHN*-overexpressing transgenic tomato plant lines (OE-2, OE-5, and OE-8) under cold and drought stress.** (A) Six-week-old wild-type and *SiDHN*-overexpressing transgenic tomato plants subjected to cold stress (4˚C and 0˚C) and allowed to recover for 3 days at 25˚C. (B) Six-week-old wild-type and *SiDHN*-overexpressing transgenic tomato plants subjected to drought stress for 28 days. Photographs were taken at 0, 16, 24, and 28 d after the treatments were initiated.

and OE-5 plants were nearly unchanged. The stems of all transgenic plants remained turgid throughout the treatment.

## Physiological changes in *SiDHN*-overexpressing transgenic tomato plants under cold and drought stress

Relative water content reflects the water retention capacity of the plants; it is used to measure water status and osmotic adjustments of the plants [35]. Before exposure to the stresses, RWC levels in the leaves of the transgenic and wild-type plants were similar (Figs 8A and 9A). By contrast, after exposure to the stresses, RWC decreased in both plant types, but the decline in the wild-type plants were considerably more accentuated than that in the transgenic plants. Under both stresses, the transgenic plants had significantly higher RWC levels than wild-type plants (P<0.01).

Malondialdehyde (MDA) is the product of lipid peroxidation caused by ROS and, combined with relative electrolyte leakage (REL), can reflect the permeability of the plasma membrane. MDA content and REL levels in the wild-type and transgenic plants were immediately elevated when exposed to cold (Fig 8B and 8C) and drought stresses (Fig 9B and 9C). Both MDA and REL in the wild-type plants were significantly higher than those the transgenic plants (P<0.01).

Proline and soluble sugar contents were also elevated in all plants under stresses, but the levels in the transgenic plants were higher than those in the wild-type plants. Under cold stress at both temperatures (4°C and 0°C), proline and soluble sugar contents of the wild-type plants were significantly lower than those of the transgenic plants (P < 0.01) (Fig 8D and 8E). Similarly, after 16 and 24 d of drought stress, proline and soluble sugar contents of the wild-type

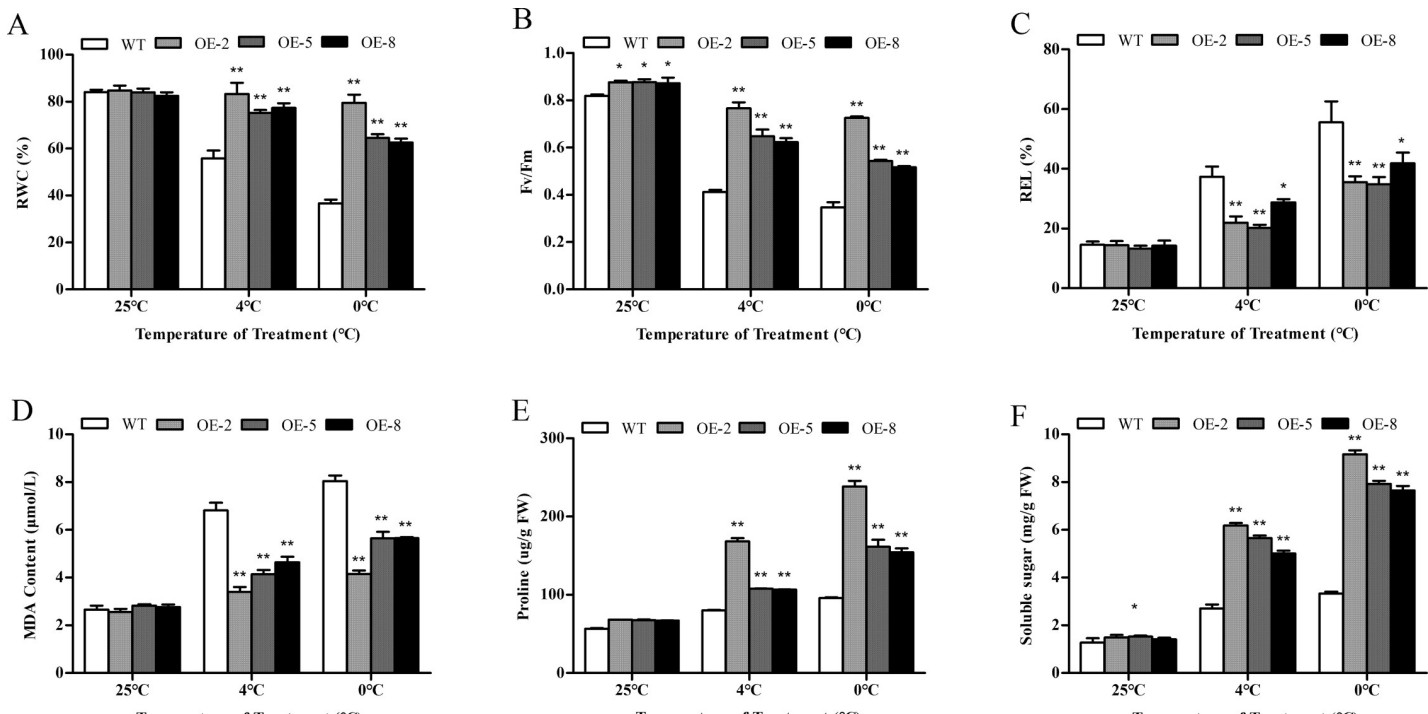

**Fig 8. Physiological changes of wild-type and *SiDHN*-overexpressing transgenic tomato plant lines (OE-2, OE-5, and OE-8) under cold stress.** (A) Relative water content. (B) $F_v/F_m$ values. (C) Relative electrolyte leakage. (D) MDA content. (E) Proline content. (F) Solute sugar content. Data are means ± SD of three replicates. Asterisk(s) indicate significant difference between the wild-type and transgenic plants: * represents P < 0.05 and ** represents P < 0.01.

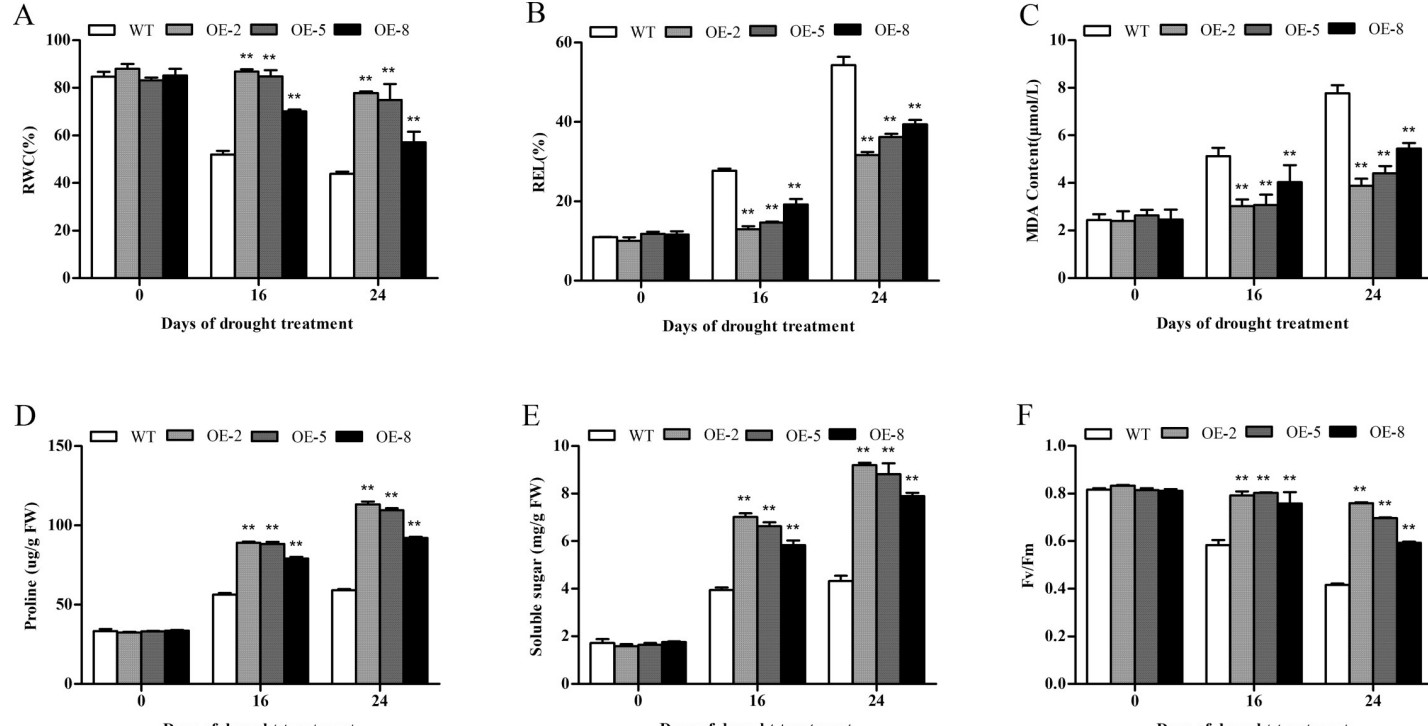

**Fig 9. Physiological changes of wild-type and *SiDHN*-overexpressing transgenic tomato plant lines (OE-2, OE-5, and OE-8) under drought stress.** (A) Relative water content. (B) Relative electrolyte leakage. (C) MDA content. (D) Proline content. (E) Solute sugar content. (F) $F_v/F_m$ values. Data are means ± SD of three replicates. Asterisk(s) indicate significant difference between the wild-type and transgenic plants: $^*$ represents $P < 0.05$ and $^{**}$ represents $P < 0.01$.

plants were significantly lower than those of the transgenic plants ($P < 0.01$) (Fig 9D and 9E). Of all transgenic plant lines, OE-2 had the highest proline and soluble sugar contents.

The maximal photochemical efficiency of photosystem II (PSII) is expressed as $F_v/F_m$. The $F_v/F_m$ values of the transgenic and wild-type plants before exposure to cold and drought stress were not significantly different (Figs 8F and 9F). But upon exposure to cold and drought, the values became significantly different ($P<0.01$); the values steadily decreased in all plants, but the decline in the wild-type plants was much more accentuated compare with that in the transgenic plants.

To study how the overexpression of *SiDHN* genes in transgenic plants under stresses affect the activities of ROS scavenging enzymes, the enzymatic activities of ascorbate peroxidase (APX), catalase (CAT), superoxide dismutase (SOD), and peroxidase (POX), as well as the contents of ROS hydrogen peroxide ($H_2O_2$) and superoxide anion ($O_2^-$) were measured. The data showed that in response to both cold and drought stresses, the enzymatic activities of APX (Figs 10A and 11A), CAT (Figs 10B and 11B), POX (Figs 10C and 11C) and SOD (Figs 10D and 11D) were up-regulated compared to those of the wild-type plants. Under drought stress, the activity levels between the two plant types were significantly different, except for CAT activities of the OE-8 plants. Under cold treatments, the levels were also significantly different among the two plant types, except for CAT activities of the OE-5 and OE-8 plant lines. Amongst the transgenic plant lines, the OE-2 had the highest expression of *SiDHN* gene and enzyme activities, whereas OE-8 had the lowest.

Before cold and drought treatments, the contents of $H_2O_2$ and $O_2^-$ in both the wild-type and transgenic plants were low and were nearly the same. Following exposure to cold stress

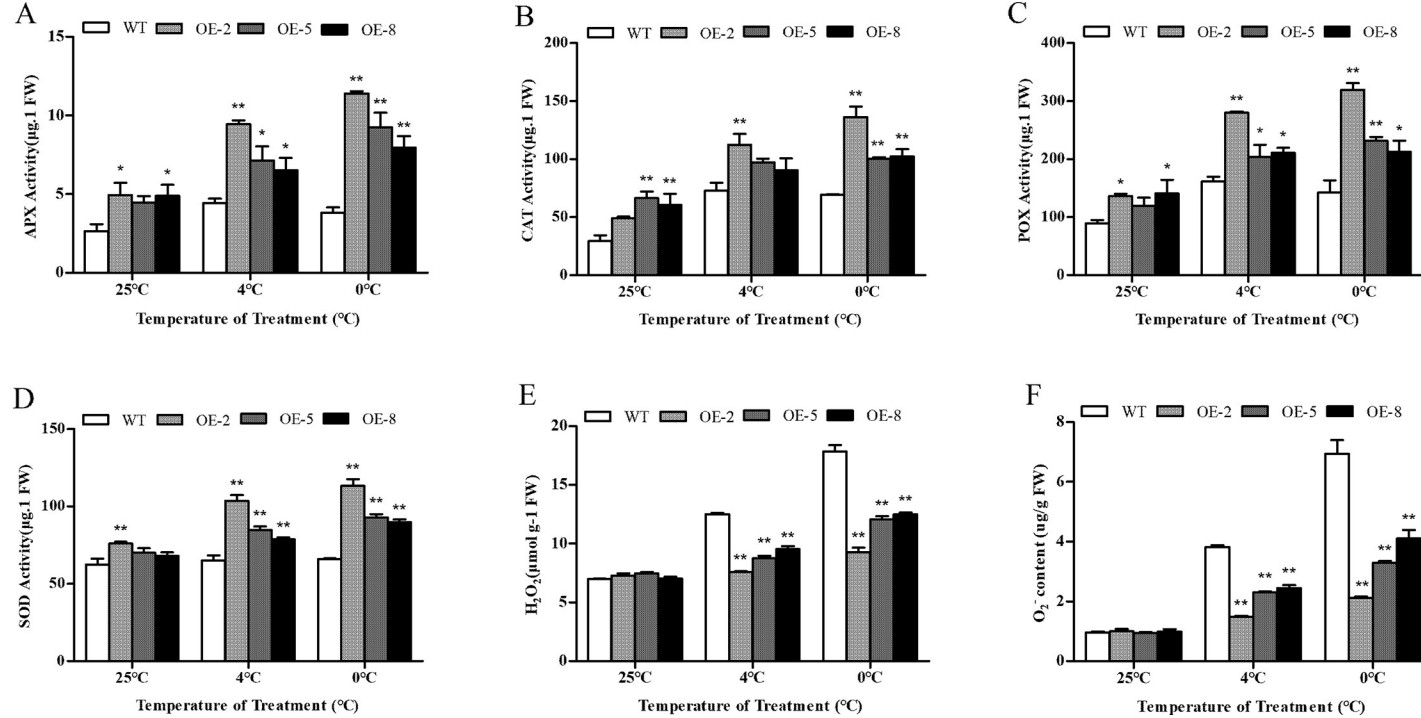

**Fig 10. Activities of antioxidant enzymes and contents of ROS in wild-type and *SiDHN*-expressing transgenic tomato plant lines (OE-2, OE-5, and OE-8) under cold stress determined at 48 h after the treatment.** (A) APX activity. (B) CAT activity. (C) POX activity. (D) SOD activity. (E) $H_2O_2$ content. (F) $O_2^-$ content. Data are means ± SD of three replicates. Asterisk(s) indicate significant difference between the wild-type and transgenic plants: * represents $P < 0.05$ and ** represents $P < 0.01$.

(Fig 10E and 10F), however, ROS levels increased, and the increase in the wild-type plants was considerably more accentuated than that in the transgenic plants. The contents of $H_2O_2$ and $O_2^-$ were significantly different between the wild-type and transgenic plants under both stresses (Fig 11E and 11F). The OE-8 plant line had the highest $H_2O_2$ and $O_2^-$ contents among all transgenic plants.

To study why transgenic plants can maintain relatively high antioxidant enzyme activity under low temperature and drought stress, we measured the expression patterns of genes *SlAPX*, *SlCAT*, *SlPOX*, and *SlSOD*, which encode the tomato's antioxidant enzymes APX, CAT, POX, and SOD, respectively, as well as genes *SlP5CS* and *SlLTP1* encoding pyrroline-5-carboxylate synthetase and lipid transfer protein 1, in both the wild-type and transgenic plants. Prior to cold treatment, the expression levels of wild-type and transgenic lines were similar; but upon treatment with low temperature, the expression levels of all plants increased, However, the expression levels of genes in transgenic lines were significantly higher than those in wild-type lines, and the expression levels increased with decreasing temperature (Fig 12). The expression of ROS-related genes *SlAPX*, *SlCAT*, *SlPOX*, *SlSOD*, *SlP5CS*, and *SlLTP1* were up-regulated in response to drought stress. The expression of these genes in the transgenic plants were significantly higher than that in the wild-type plants, even after 16 d. Amongst the transgenic plant lines, OE-2 had the highest expression levels, whereas OE-8 had the lowest (Fig 13). The expression levels of genes *SlAPX*, *SlCAT*, *SlPOX*, and *SlSOD* in plants under stress were highly similar to those of antioxidant enzymes, which suggests that the relative level of antioxidant enzymes may be determined by the level of their genes. The results indicate that overexpression of *SiDHN* may be able to improve the expression of antioxidant related genes, thus increasing the activity of the corresponding antioxidant enzymes, improving the ROS

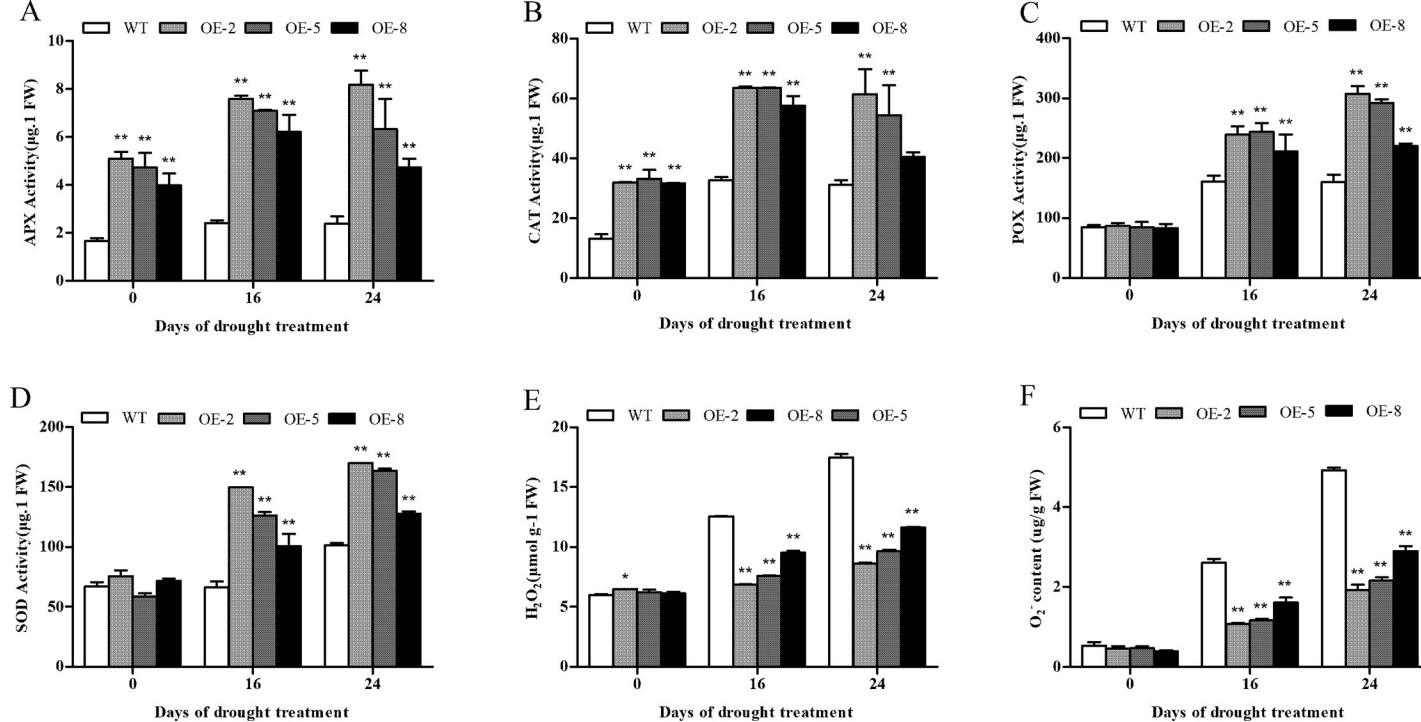

**Fig 11. Activities of antioxidant enzymes and accumulation of ROS in wild-type and *SiDHN*-overexpressing transgenic tomato plant lines (OE-2, OE-5 and OE-8) under drought stress.** (A) APX activity. (B) CAT activity. (C) POX activity, (D) SOD activity. (E) $H_2O_2$ content. (F) $O_2^-$ content. Data are means ± SD of three replicates. Asterisk(s) indicate significant difference between the wild-type and transgenic plants: * represents P < 0.05 and ** represents P < 0.01.

clearance, and in turn reducing the cell membrane damage caused by low temperature- and drought-induced oxidative stress.

## Discussion

The growth, development, and productivity of crops are affected by elevated levels of ROS arising from abiotic stress, including cold temperatures and drought. In response to these stresses, a number of regulatory and/or protective proteins are produced to stabilize membrane phospholipids, membrane proteins, and cytoplasmic proteins [52], to maintain hydrophobic interactions and ion homeostasis, and to scavenge ROS [53]. Among these proteins, dehydrins are known as the characteristic stress proteins produced during abiotic stress [54].

In this study, we first investigated dehydrin gene isolated from *S. involucrate* (*SiDHN*) and its expression levels in response to cold and drought stress. Our sequence analysis indicated that SiDHN protein is highly hydrophilic and belongs to the dehydrin $YK_2$-type (Fig 1). According to Qiu et al. and Guo et al. [55, 22], such sequence properties are plants' essential ability to reduce water loss when they are under stress conditions. The SiDHN protein also contains over 78% of disordered structure, suggesting that it is an intrinsically disordered protein (IDP).

We further found that the expression of *SiDHN* gene was up-regulated in *S. involucrata* treated with cold and drought stress (Fig 2), confirming that the *SiDHN* gene is involved in the plant's response to abiotic stress [54]. Tissue-specific profiles also showed that *SiDHN* was expressed in the roots, stems, and leaves of *S. involucrate*, and the levels in leaves were highest among all three tissues, suggesting that leaves may be the first targets of abiotic stress.

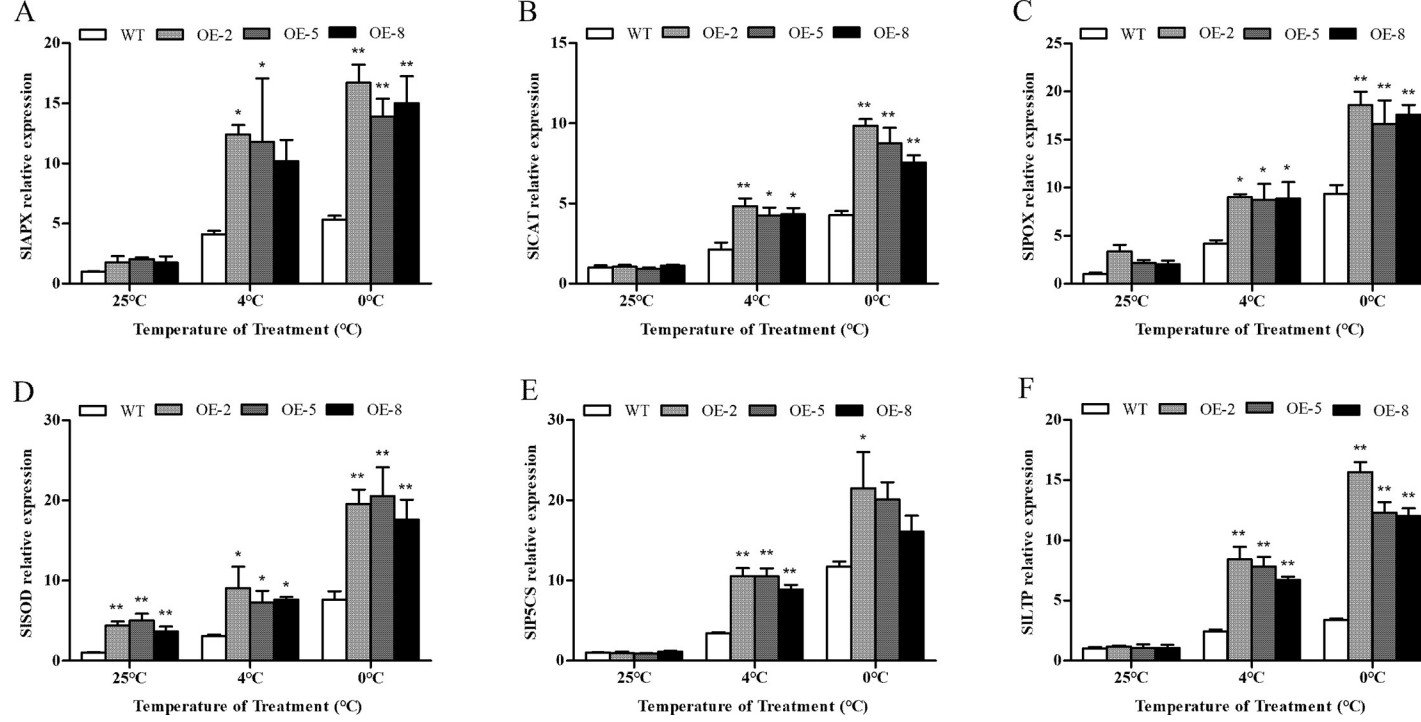

**Fig 12. Relative gene expression of *SlAPX*, *SlCAT*, *SlPOX*, *SlSOD*, *SLP5CS*, and *SlLTP* in *SiDHN*-overexpressing transgenic tomato plant lines (OE-2, OE-5, and OE-8) under cold stress (4°C and 0°C).** Data are means ± SD of three replicates. Asterisk(s) indicate significant difference between the wild-type and transgenic plants: * represents P < 0.05 and ** represents P < 0.01.

We then determined whether the *SiDHN* gene could help other plants protect against abiotic stress, and investigate the mechanism in which it exerts the effects. We transformed the *SiDHN* gene isolated from *S. involucrata* into wild-type tomato plants to produce transgenic tomato plants. We then observed agronomic traits, morphological changes and physiological changes, and determined changes of chlorophyll content, various gene expressions and enzyme activities of the transgenic plants. We found that the leaves of transgenic plants had denser and darker color (Fig 4) and contained higher Chl a and Chl b contents in the transgenic tomato plants (Fig 6) compared with the wild-type plants. Chlorophylls are the core components of pigment-protein complexes and play a major role in photosynthesis. They also help degrade ROS. Chlorophylls localize in chloroplasts, which are a major source of ROS largely produced during cold and drought stress; thus, when plants are exposed to cold and drought stress, chlorophylls are largely produced to help degrade ROS [56]. As a result, chlorophyll contents increased. The carotenoid (Car) content in the two of the three plant lines generated (OE-2 and OE-8) was also higher compared to that in the wild-type plants. The heights of transgenic tomato plants were shorter than those of he wild-type plants, in addition longer length of root, higher fresh and dry weight of root, and higher stem diameters (Fig 5), which indicates that the overexpression of *SiDHN* gene affects the growth and development of transgenic tomato plants.

Cold and drought stress generally leads to the decline of RWC. The decline in RWC in the wild-type plants was more accentuated compared to that in the transgenic plants (Figs 8A and 9A), suggesting that *SiDHN* gene may play a role in protecting the transgenic tomato plants against cold and drought stress. Similar observations have been reported, in which the overexpression of wheat *DHN-5* and Musa *DHN-1* in transgenic *Arabidopsis thaliana* under drought

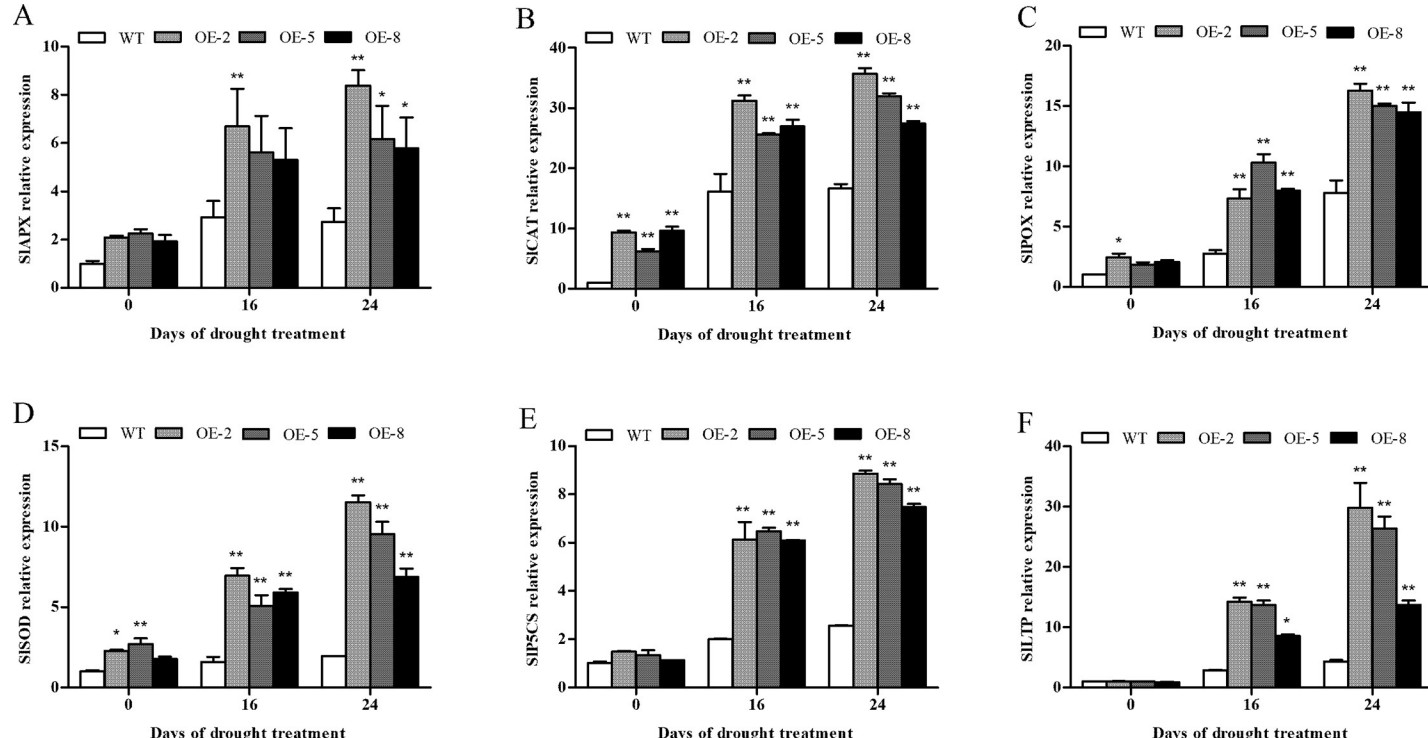

**Fig 13. Relative expression of *SlAPX*, *SlCAT*, *SlPOX*, *SlSOD*, *SLP5CS* and *SlLTP* genes in *SiDHN*-overexpressing transgenic tomato plants lines (OE-2, OE-5, and OE-8) under drought conditions for 16 and 24 d.** Data are means ± SD of three replicates. Asterisk(s) indicate significant difference between the wild-type and transgenic plants: * represents P < 0.05 and ** represents P < 0.01.

stress was found to help alleviate the declined RWC [57]. Other studies have also shown that dehydrin can replace water molecules and combine with negatively charged biofilm liposomes to maintain the stability of biofilm [56]; thus, when combines with water molecules, they can reduce the water loss rate in leaves during water deficit stress [57, 58].

ROS produced when plants are under oxidative stress leads to higher production of MDA due to lipid peroxidation of cell membranes. Liu et al have previously shown that the heterologous expression of the maize *ZmDHN13* gene causes the reduction of in MDA content in tobacco subjected to oxidative stress [20]. This is similar to our results, which showed that MDA contents in the transgenic plants under both cold and drought stress were lower compared with those in the wild-type plants (Figs 8D and 9D). The REL in transgenic lines were also higher, implying that overexpression of *SiDHN* could increase plants' ability to protect against membrane damage (Figs 8C and 9C).

Proline and soluble sugar contents in the transgenic plants were also increased compared with those in the wild-type plants (Figs 8 and 9). Proline is considered as a membrane stabilizer, an osmoprotectant, and a scavenger of free radicals [59]. In addition, aside from being a source of energy and carbon, soluble sugar accumulation has been reported to improve osmoregulation, protect proteins, and maintain photosynthesis during low water stress [2].

ROS is removed through the ROS-scavenging enzymes [60]. The superoxide dismutase (SOD) first catalyzes the dismutation of $O_2^-$ in $H_2O_2$, and the $H_2O_2$ is subsequently converted to $H_2O$ by ascorbate peroxidase (APX), peroxidase (POX), and catalase (CAT). Studies have shown that increases in SOD, POX, and CAT activities enhance stress tolerance of dehydrin-overexpressing plants during oxidative stress [18, 20]. Our study showed that under stress,

SOD activity in transgenic plant lines was elevated compared to that in the wild-type plants (Fig 10D and 11D). We also observed that the activity of other antioxidant enzymes also increased, suggesting that after the catalysis of $O_2^-$ into $H_2O_2$ by SOD, POX and APX can reduce $H_2O_2$ into water. CAT also appears to help maintain ROS homeostasis, according to our results; however, aside from the OE-2 plants, the CAT activity only slightly increased in activity at 4˚C (Fig 10B). Similar results were observed in the study on Sorghum bicolor dehydrins by Hadler et al in which CAT activity was found to be lower, which suggests that CAT activity may depend on $H_2O_2$ concentrations [18]. Mittler has previously reported that POX and APX can exert their functions even at low ROS levels, which indicates that both enzymes are responsible for modulation of $H_2O_2$ levels through necessary signaling events; whereas, CAT is mainly involved in preventing $H_2O_2^-$ induced cellular damage by removing its excess [3]. We also demonstrated that the expression of *SlSOD*, *SlAPX*, *SlPOX*, and *SlCAT* genes were up-regulated during cold and drought stress (Figs 12 and 13). Collectively, these results suggest that *SiDHN* had a regulatory role in mobilizing antioxidant enzymes to catalyze the ROS, lowering its level to a non-toxic level, minimizing oxidative damage to the plants.

The expression of gene encoding pyrroline-5-carboxylate synthetase (*SlP5CS*) and lipid transfer protein (*SlLTP1*) were also enhanced upon stress treatments. *SlP5CS* is a key proline synthetase gene and, given the elevated levels of proline previously described, its increase is unsurprising. LTPs are involved in membrane biogenesis and the transport of phospholipids [61–63]. Although their mechanisms are not well understood, high LTP expression has been reported to increase plant tolerance to cold, drought and elevated $H_2O_2$ levels [64–65]. These results indicate that *SiDHN* modulates the expression of genes involved in stress signaling pathways, which may be one the mechanisms of enhancing tolerance to cold and drought stress of plants.

Taken together, we observed that *SiDHN*, a downstream effector of plant responses to cold and drought stress, could improve stress responses of transgenic tomato plants by maintaining the integrity of the cell membrane, reducing chlorophyll photo-oxidation and reactive oxygen species accumulation, and improving antioxidant enzyme activity and photochemical electron transfer efficiency. Tolerance to drought and cold stress is important plant traits; therefore, *SiDHN* is a promising candidate gene for genetic engineering of plants to have high drought and cold stress tolerance, which could lead to large economic benefits for agricultural production. Nonetheless, further study on functions and activity of *SiDHN* in *S. involucrata* should be conducted.

## Author Contributions

**Conceptualization:** Jianbo Zhu.

**Data curation:** Li Zhang.

**Formal analysis:** Li Zhang, Aiying Wang.

**Methodology:** Li Zhang.

**Project administration:** Xinyong Guo, Xiaozhen Wang.

**Software:** Minhuan Zhang.

**Supervision:** Xinyong Guo, Xiaozhen Wang, Yuxin Xi.

**Validation:** Minhuan Zhang, Aiying Wang.

**Visualization:** Yuxin Xi, Aiying Wang.

**Writing – original draft:** Xinyong Guo.

**Writing – review & editing:** Xinyong Guo, Jianbo Zhu.

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
