## [Decision Letter · Decision Letter 0]

6 Aug 2019

PONE-D-19-17695

Overexpression of Saussurea involucrata Dehydrin Gene SiDHN Promotes Cold and Drought Tolerance in Transgenic Tomato Plants

PLOS ONE

Dear %Dr. Jianbo Zhu%,

Thank you for submitting your manuscript to PLOS ONE. After careful consideration, we feel that it has merit but does not fully meet PLOS ONE’s publication criteria as it currently stands. Therefore, we invite you to submit a revised version of the manuscript that addresses the points raised during the review process.

We would appreciate receiving your revised manuscript by %30th September, 2019%. To enhance the reproducibility of your results, we recommend that if applicable you deposit your laboratory protocols in protocols.io, where a protocol can be assigned its own identifier (DOI) such that it can be cited independently in the future. For instructions see: http://journals.plos.org/plosone/s/submission-guidelines#loc-laboratory-protocols

We look forward to receiving your revised manuscript.

Kind regards,

T. R. Ganapathi, Ph.D.

Academic Editor

PLOS ONE

Journal Requirements:

1. We noticed you have some minor occurrence of overlapping text with the following previous publication(s), which needs to be addressed:

https://www.sciencedirect.com/science/article/abs/pii/S0098847218310566?via%3Dihub

https://link.springer.com/article/10.1007%2Fs00344-017-9718-2

https://www.frontiersin.org/articles/10.3389/fpls.2017.01659/full

In your revision ensure you cite all your sources (including your own works), and quote or rephrase any duplicated text outside the methods section. Further consideration is dependent on these concerns being addressed.

2. Please amend your list of authors on the manuscript to ensure that each author is linked to an affiliation. Authors’ affiliations should reflect the institution where the work was done (if authors moved subsequently, you can also list the new affiliation stating “current affiliation:….” as necessary).

Additional Editor Comments (if provided):

The manuscript requires major revisions based on the reviewers comments for considering it for publication.

Reviewers' comments:

Reviewer's Responses to Questions

**Comments to the Author**

1. Is the manuscript technically sound, and do the data support the conclusions?

Reviewer #1: Partly

Reviewer #2: Partly

2. Has the statistical analysis been performed appropriately and rigorously? 

Reviewer #1: Yes

Reviewer #2: Yes

3. Have the authors made all data underlying the findings in their manuscript fully available?

Reviewer #1: Yes

Reviewer #2: Yes

4. Is the manuscript presented in an intelligible fashion and written in standard English?

Reviewer #1: No

Reviewer #2: No

5. Review Comments to the Author

Reviewer #1: Title of Manuscript: Overexpression of Saussurea involucrata Dehydrin Gene SiDHN Promotes Cold and Drought Tolerance in Transgenic Tomato Plants

In the present manuscript, authors reported mechanisms underlying cold and drought tolerance in transgenic tomato through over expression of Saussurea involucrata dehydrin gene SiDHN. The study concluded that the engineered stress tolerance was associated with better osmotic adjustment and reduced oxidative damage.

The manuscript appears to be of sufficient scientific interest and originality in its technical content to merit publication. However, it is poorly drafted and need to be re-written in consultation with the language expert. Methodology followed is sufficient to draw the conclusions, however several important details are missing. The data is analyzed statistically. Obtained results have been discussed in light of the earlier scientific reports in the area.

Section wise shortcomings are mentioned below, which need to be addressed by the authors before acceptance of the manuscript.

1. Abstract: The term “tolerance” appears to be more appropriate instead of “resistance” for improved performances against abiotic stresses. Conclusion need to be written more specifically.

2. Introduction: Statements need to be supported with published scientific literature. For example please refer line no 92-95. Necessity of the research work also needs to be clearly stated.

3. Materials and methods: Plant lets of S. involucrate were regenerated through tissue culture, therefore use term “plantlets” instead of “seedlings”. Details are missing on explants, medium etc. for tissue culture of the plant. The purpose of using tissue culture raised plants need to be mentioned in the manuscript.

4. Materials and methods: Sequence analysis of SiDHN: Details on primer designing and sequencing of the PCR amplicon are missing.

5. Materials and methods: Plant Transformation: Details are missing on cloning of the gene and Agrobacterium transformation. Important details are missing on genetic transformation of tomato such as explant, co-cultivation, regeneration etc. Details are missing on RNA isolation, RT etc.

6. Materials and methods: Stress tolerance: How many plants were used for treatment? The stress treatment for short period of 2h imposes shock to the plants. Why longer treatments were not given to assess the stress tolerance and real response of the plants? Details pertaining to the bio-chemical analyses are missing. It is important to mention the biological and technical replicates for the analyses.

7. Results: Transgenic confirmation with mere kanamycin selection and PCR may not be sufficient. Statistical significance, if any, for the parameters among the transgenic and wild types need to be clearly stated rather than just mentioning the higher or lower change. The reason behind studying the transcript expression of few genes is not clearly stated. What was the criterion for selection of the genes? Whether, the PCR amplicons for the selected genes were confirmed by sequencing? What was the relation between the transcript level and activity of the coded anti-oxidant enzymes?

8. Discussion: Author stated that “expression of SiDHN was induced in S. involucrata by cold temperatures and drought (Fig 2), confirming that the gene participates in the plant’s defenses to abiotic stress”. However, it is necessary to understand that mere change in expression level of any gene does not confirm its role in plant defence, functional validation is required in support of such statement.

Recommendation: Major revision

Reviewer #2: The manuscript entitled “Overexpression of Saussurea involucrata Dehydrin Gene SiDHN Promotes Cold and Drought Tolerance in Transgenic Tomato Plants” describes the effect of overexpression of SiDHN in improving cold and drought stress tolerance in tomato. The study is a continuation of their earlier work (Plant Sci. 2017; 256:160–169). It is interesting but lacks detailed discussion, unexplained and/or mechanistic view of the observations. Several papers are published on dehydrins and their role of in improving abiotic stress tolerance in plants. I have the following comments which may help to improve the manuscript.

1- Introduction, material and methods and results are well covered, but the discussion is poorly written. They must explain/discuss the findings in the light of available literature.

2- Resolution of most of the figures is poor.

3- Manuscript requires extensive editing (scientific as well as grammatical). There are several errors across MS. A few are:

Line 45: Use specific word than slightly.

Line 50: within the transgenic?

Line 73: tied to the acquisition?

Line 192: change 13000 rpm to x g

Youngest fully expanded leaves?

Line 120-121: gene and scientific name in italics

Line 193: Supernatant fluid?

4- Line 40-45: Write shorter and clear sentences.

5- Line 83-84: References

6- Line 137: Whether author are sure they have used just 80µg/L kanamycin for selection?

7- For the freezing stress, authors gave treatment for 4⁰C/2h and 0⁰C/2h. Why longer duration not selected?

8- Why two methods were used for CAT & POX estimation?

9- What is the relevance of ref. 35?

10- Conditions for the enzyme analysis (4⁰C/2h and 0⁰C/2h) and gene expression analysis (4°C chamber for 48 h and then kept at 0°C for 48 h) were different. Why?

11- Line 366: Authors claim that over-expression of SiDHN does not have any visible effect on transgenic plants. It is contradictory to what they mentioned in 326-355. They must explain why transgenic plants have higher chlorophyll content and a shorter height than control plants? WT-plants does not appear healthy. Under normal growth conditions, why leaves will have less chlorophyll. Why will dehydrin give advantage to transgenic plants under normal condition? They must explain this.

12- 213-214: Dehydrated on filter paper: Is this a standard method for drought stress? To me, it’s not convincing as there can be many variables.

13- 235: For real-time PCR, whether the author followed the MIQE guidelines? If yes, the same should be mentioned.

14- Whether MEANS in Fig. 2 were compared (statistics)? Same is not reflected.

15- There was a continuous increase in SiDHN expression under cold stress during 0-48h, unlike in drought. Why was the expression of SiDHN not monitored beyond 48h?

16- 316-319: SiDHN from S. involucrata was over-expressed (heterologous) in tomato plants. So there should not be any expression of the gene in WT-plants. But in Fig. 3A, a certain level of expression is shown. Also, they should explain if there is no expression in WT-plants, how 292,241, 139-fold mRNA levels than WT-plants was observed?

17- They have not included vector-control plants for any of their studies. Why?

18- Fig. 7: Looking at fig, it's difficult to believe uniformity among control and transgenic plants. To me, a lot of variation is visible even on day 0 in plant phenotype between even cold/drought stress experiments.

19- 454-455: Fig. 10E, F & 11 E,F should be 9 EF & 12 E,F.

20- Why levels of H2O2 and O2- were not monitored on 0, 24, 48h. That would have been one of the important aspects to link with the status of downstream antioxidant enzymes. Levels of H2O2 and O2 should be included in duration dependent ie. 0, 24, 48h to correlate the things with levels of enzymes.

21- Line 524: How would the author explain the higher levels of expression of SiDHN in leaves?

22- Fig: 7, whether the recovery of plants was observed for a longer duration. What were the results post 3days recovery?

23- Levels of SiDHN was found highest in OE8, but levels of most of the antioxidant enzymes including tolerance and survival under stress were observed in OE2 (Recovery). How would the author explain it? Similarly, in the case of expression of analysis of APX, SOD, POX (Fig. 10) shows an increase in their levels with duration. But it’s difficult to understand unless it is shown that H2O2/O2- levels also increased.

6. PLOS authors have the option to publish the peer review history of their article (what does this mean?). If published, this will include your full peer review and any attached files.

Reviewer #1: Yes: Dr Patade Vikas Yadav

Reviewer #2: No

---

## [Author Response · Author response to Decision Letter 0]

10 Sep 2019

Comments from Reviewer 1

Comment 1: Abstract: The term “tolerance” appears to be more appropriate instead of “resistance” for improved performances against abiotic stresses. Conclusion need to be written more specifically.

Response: Thank you for pointing this out, and we agree that these should be changed. Accordingly, we have changed the term and improved the conclusion in the abstract. Please refer to lines 38 and 50 – 55. 

Comment 2: Introduction: Statements need to be supported with published scientific literature. For example please refer line no 92-95. Necessity of the research work also needs to be clearly stated.

Response: Thank you very much for pointing out. We have ensured that all statements are supported by the published literature. We have also edited and rearranged the statements so that the necessity of the research work is more clearly stated, which can be found in lines 82 – 95.

Comment 3: Materials and methods: Plant lets of S. involucrate were regenerated through tissue culture, therefore use term “plantlets” instead of “seedlings”. Details are missing on explants, medium etc. for tissue culture of the plant. The purpose of using tissue culture raised plants need to be mentioned in the manuscript.

Response: We agree with the reviewer on these points. Accordingly, we have changed from ‘seedlings’ to ‘plantlets’, where we find necessary, and have provided the details on explants, medium, purpose of using tissues culture to raise plants. We have also included other details necessary for the understanding of the section. We invite the reviewer to check the ‘Plant materials and growth conditions’ section in lines 111 – 122. 

Comment 4: Materials and methods: Sequence analysis of SiDHN: Details on primer designing and sequencing of the PCR amplicon are missing.

Response: We agree with you on this point. We have incorporated the details on how the primer was designed at the bottom of Table 1, as well as on the sequencing of PCR amplicon, which was done after plasmid extraction (Lines 210 – 220). 

Comment 5: Materials and methods: Plant Transformation: Details are missing on cloning of the gene and Agrobacterium transformation. Important details are missing on genetic transformation of tomato such as explant, co-cultivation, regeneration etc. Details are missing on RNA isolation, RT etc.

Response: Thank you so much for pointing this out. We agree with you and have incorporated new details on cloning and Agrobacterium transformation, as well as of transformation of tomato. We invite you to check the ‘Plant transformation and transgenic tomato identification’ section (Lines 210 – 237), which has been largely edited.

Comment 6: Materials and methods: Stress tolerance: How many plants were used for treatment? The stress treatment for short period of 2h imposes shock to the plants. Why longer treatments were not given to assess the stress tolerance and real response of the plants? Details pertaining to the bio-chemical analyses are missing. It is important to mention the biological and technical replicates for the analyses.

Response: Thank you for this comment. Please find a point-by-point response below.

Response to “How many plants were used for treatment?”: We used 3 wild-type and 9 T2 transgenic tomato plants, both were ten weeks old. We have also added the corresponding details in the manuscript. Please refer to lines 141 – 142. 

Response to “The stress treatment for short period of 2h imposes shock to the plants”: We apologize for this error. The stress treatment was, in fact, carried out for 48 h (or 2 d), and we have changed the content accordingly. Please see line 143. 

Response to “Why longer treatments were not given to assess the stress tolerance and real response of the plants?”: Thank you for pointing this out. We did, in fact, carry out the treatment at a longer period, at which the stress tolerance appeared to be deteriorated. We decided to not include the data in the manuscript; however, we have included some details in the manuscript (results section) to indicate the finding. Please refer to lines 396 – 397.

Response to “Details pertaining to the bio-chemical analyses are missing”: We apologize for this unclarity. We have included more details pertaining the biochemical analyses, which can be found at the beginning of each paragraph of the sections ‘Measurement of chlorophyll pigment content’ (lines 240 – 253) and ‘Determination of morphological and physiological traits’ (lines 268 – 270, lines 276 – 277, lines 283 – 285, lines 293 – 297, lines 304 – 306).

Response to “It is important to mention the biological and technical replicates for the analyses”: We have included sentences stating that “Each experiment was carried out in triplicate, and three replicate samples were prepared in parallel” in all sections, where we find appropriate, such as in lines 323 – 324.

Comment 7: Results: Transgenic confirmation with mere kanamycin selection and PCR may not be sufficient. Statistical significance, if any, for the parameters among the transgenic and wild types need to be clearly stated rather than just mentioning the higher or lower change. The reason behind studying the transcript expression of few genes is not clearly stated. What was the criterion for selection of the genes? Whether, the PCR amplicons for the selected genes were confirmed by sequencing? What was the relation between the transcript level and activity of the coded anti-oxidant enzymes?

Response: Thank you for pointing this out. We carried out the study following a number of publications that have used the same selection method (e.g., reference [22]), which have been proven to be sufficient. We have edited the sentences, incorporating the term ‘significantly different’ or ‘significantly higher or lower’ in various areas of the manuscript, where we find appropriate. We have also largely modified the content to address the concerns above, but made throughout different areas on the manuscript. We invite the reviewer to recheck out new content, such as lines 578 – 582, lines 548 – 556, and the discussion section lines 674 – 706. 

Comment 8: Discussion: Author stated that “expression of SiDHN was induced in S. involucrata by cold temperatures and drought (Fig 2), confirming that the gene participates in the plant’s defenses to abiotic stress”. However, it is necessary to understand that mere change in expression level of any gene does not confirm its role in plant defence, functional validation is required in support of such statement.

Response: Thank you for this comment. In fact, the participation of SiDHN in the plant’s defenses to abiotic stress has been reported previously, as can be found in, for example, reference [22] and [57], and our study confirmed such participation.

Comments from Reviewer 2

Comment 1: Introduction, material and methods and results are well covered, but the discussion is poorly written. They must explain/discuss the findings in the light of available literature.

Response: Thank you for pointing this out. We have largely edited and rearranged the entire discussion section so that it is presented in the way that the reviewer suggested. We do hope that it is sufficient. We invite you to check the ‘Discussion’ section in lines 609 – 716.

Comment 2: Resolution of most of the figures is poor.

Response: Thank you for pointing this out. We have increased the resolution of the figures, which we find most important for the comparison. 

Comment 3: Manuscript requires extensive editing (scientific as well as grammatical). There are several errors across MS. A few are:

Line 45: Use specific word than slightly.

Line 50: within the transgenic?

Line 73: tied to the acquisition?

Line 73: tied to the acquisition?

Line 192: change 13000 rpm to x g

Youngest fully expanded leaves?

Line 120-121: gene and scientific name in italics

Line 193: Supernatant fluid?

Response: We appreciate the reviewer for pointing this out. We have corrected/changed the errors as recommended in the comment, as well as in the entire manuscript as we spotted them. Please refer the specific response below.

Response to “Line 45: Use specific word than slightly”: We have changed the sentence to avoid using the word ‘slightly’ in this sentence (lines 45 – 46), as well as in all other sentences, where applicable. In many cases, we have instead incorporated the term ‘significantly’ (based on the statistical analysis).

Response to “Line 50: within the transgenic?”: We have changed the expression to ‘in the transgenic plants’ (lines 46 – 47). We have also used similar expression throughout the manuscript, where we find appropriate.

Response to “Line 73: tied to the acquisition?”: We have changed this expression to ‘are closely linked with’. Please refer to line 73.

Response to “Line 192: change 13000 rpm to x g”: We have changed rpm to xg throughout the manuscript. The original line 192 has now been moved to a new line 332. 

Response to “Youngest fully expanded leaves?”: The intent of this phrase is to indicate that the leaves are young but are fully-grown. Thus, we have changed this to ‘young, fully-grown leaves’. Please refer to lines 287 – 288.

Response to “Line 120-121: gene and scientific name in italics”: We have italicized the gene and scientific names accordingly throughout the manuscript. 

Response to “Line 193: Supernatant fluid?”: We have changed this to ‘the supernatant’, and have also rearranged the sentence. Please refer to line 309.

Comment 4: Line 40-45: Write shorter and clear sentences.

Response: We have removed repetition and redundancy in these sentences and changed them to: 

‘We observed that in response to stresses, the SiDHN transgenic tomato plants had increased contents of chlorophyll a and b, carotenoid and relative water content compared with wild-type plants. They also had higher maximal photochemical efficiency of photosystem II and accumulated more proline and soluble sugar. Compared to those wild-type plants, malondialdehyde content and relative electron leakage in transgenic plants were not significantly increased, and H2O2 and O2- contents in transgenic tomato plants were significantly decreased.’ 

Please refer to lines 40 – 47. 

Comment 5: Line 83-84: References

Response: Thank you for pointing this out. We have recounted the references and have inserted the appropriate reference numbers accordingly. Please refer to lines 82 – 84.

Comment 6: Line 137: Whether author are sure they have used just 80µg/L kanamycin for selection?

Response: Thank you for this question. We carried out the experiment according to the previously published article in which 80µg/L kanamycin was used for selection. We also carried out RT-PCR to confirm that the gene has been transformed.

Comment 7: For the freezing stress, authors gave treatment for 4⁰C/2h and 0⁰C/2h. Why longer duration not selected?

Response: We apologize for this mistake. We, in fact, conducted the treatment at each temperature for 48 h and have accordingly corrected the detail. We also carried out the treatments at a longer time period, but decided not to include it in the data. We have, however, stated in the manuscript (result section) that the tolerance was decreased at a period longer than 48 h, which can be found in lines 395 – 397.

Comment 8: Why two methods were used for CAT & POX estimation?

Response: We cited two references (for each estimation) that describe the same method. We have reduced the number of reference and cited to the one published in earlier year. Please refer to lines 336 and 339.

Comment 9: What is the relevance of ref. 35?

Response: This appears to be a mistake on our end, and we apologize for that. We, however, have made extensive changes in the manuscript, whereby reference [35] is now no longer relevant. We invite you to check the edited content, especially for the section ‘Plant treatments’ (lines 132 – 151), in which the reference [35] was originally cited.

Comment 10: Conditions for the enzyme analysis (4⁰C/2h and 0⁰C/2h) and gene expression analysis (4°C chamber for 48 h and then kept at 0°C for 48 h) were different. Why?

Response: We apologize for the error. This should, in fact, be 2 d or 48 h. However, I have changed it to 48 h for consistency.

Comment 11: Line 366: Authors claim that over-expression of SiDHN does not have any visible effect on transgenic plants. It is contradictory to what they mentioned in 326-355. They must explain why transgenic plants have higher chlorophyll content and a shorter height than control plants? WT-plants does not appear healthy. Under normal growth conditions, why leaves will have less chlorophyll. Why will dehydrin give advantage to transgenic plants under normal condition? They must explain this.

Response: We apologize for this mistake. The overexpression of SiDHN did, in fact, affect the phenotypes of the transgenic plants. Thus, we have made some revisions regarding this issue both in the results and the discussion sections, which should cover the questions stated above. We invite the reviewer to re-check the revised contents; for example, in the ‘results’ sections lines 470 – 471, and in the ‘discussion’ section lines 632 – 648. 

Comment 12: 213-214: Dehydrated on filter paper: Is this a standard method for drought stress? To me, it’s not convincing as there can be many variables.

Response: Thank you for this comment. We, in fact, carried out this experiment following the method described in Zhu et al [23], which has been proven to be sufficient. 

Comment 13: 235: For real-time PCR, whether the author followed the MIQE guidelines? If yes, the same should be mentioned.

Response: Thank you for pointing this out. Yes, we did follow the MIQE guidelines and accordingly, have included the statement in the manuscript. Please refer to line 161.

Comment 14: Whether MEANS in Fig. 2 were compared (statistics)? Same is not reflected.

Response: Thank you for this comment. We did all comparisons statistically and have included the details in the description of figure 2, as well as others where we find necessary for the better understanding of the content. 

Comment 15: There was a continuous increase in SiDHN expression under cold stress during 0-48h, unlike in drought. Why was the expression of SiDHN not monitored beyond 48h?

Response: We, in fact, carried out the treatment at each temperature beyond 48 h but decided not to include it in the data. We have, however, stated the finding in the manuscript (result section), which can be found in lines 396 – 397.

Comment 16: 316-319: SiDHN from S. involucrata was over-expressed (heterologous) in tomato plants. So there should not be any expression of the gene in WT-plants. But in Fig. 3A, a certain level of expression is shown. Also, they should explain if there is no expression in WT-plants, how 292,241, 139-fold mRNA levels than WT-plants was observed?

Response: Thank you for this comment. Yes, there was indeed no expression of the SiDHN gene in the wild-type potato plants. The comparison was based on the assumption set in ‘Statistical analysis’ section, which states that ‘the expression level at 0 h (control time point) was defined as 1.0’. However, we decided to not include these numbers in the manuscript, rather we stated that ‘Three independent homozygous lines (OE-2, OE-5, and OE-8) that had higher expression levels of SiDHN were selected for further studies (Fig 3).’ Please refer to lines 422 – 423. 

Comment 17: They have not included vector-control plants for any of their studies. Why?

Response: Thank you for pointing this out. We carried out the experiment following the previous publication on SiDHN (e.g. reference [22]); thus, we decided that not including vector-control plants should be appropriate. And we believe that the results should provide insights into the understanding of SiDHN gene expression to some extent. 

Comment 18: Fig. 7: Looking at fig, it's difficult to believe uniformity among control and transgenic plants. To me, a lot of variation is visible even on day 0 in plant phenotype between even cold/drought stress experiments.

Response: We agree with the reviewer on this and have changed/edited some sentences so that the content more clearly presents such differences. We invite the reviewer to re-check the section ‘Morphological changes of SiDHN-overexpressing transgenic tomato plants under cold and drought stress’ (Lines 470 – 471).

Comment 19: 454-455: Fig. 10E, F & 11 E,F should be 9 EF & 12 E,F.

Response: Thank you very much for pointing out this error. We have changed the content accordingly. Please see lines 558 – 559. 

Comment 20: Why levels of H2O2 and O2- were not monitored on 0, 24, 48h. That would have been one of the important aspects to link with the status of downstream antioxidant enzymes. Levels of H2O2 and O2 should be included in duration dependent ie. 0, 24, 48h to correlate the things with levels of enzymes.

Response: We decided to determine the levels of H2O2 and O2- in correspondent to the enzyme activity (Fig. 9), which was determined at the end of the treatment (48 h). We think that in our future study, we should determine such levels at various times as the reviewer suggested. In addition, because the gene expression does not necessarily determine the enzyme activity; therefore, we measured the levels of the levels of H2O2 and O2- only when the enzyme activity was measured so that we can better correlate the results. 

Comment 21: Line 524: How would the author explain the higher levels of expression of SiDHN in leaves?

Response: There have been only a few studies on the effects of SiDHN overexpression on plant growth and development. For example, Nylander et al. [55] believed the expression of the dehydrin gene may be related to its function and can also be affected by different stages of plant development under stress. Stupnikova et al. [56] proposed that plants have different resilience at different developmental stages: the expression of dehydrin is high when they are under high resilience. These studies have also shown that the expressions of many dehydrins are specific to tissues, and the growth conditions can also affects the expression levels. The studies, however, have not focused on the expression in different plant tissues. Nonetheless, based on out results, we believe that the higher level of expression in leaves may be due to high resilience in leaves, which may be the first target of stress. We have accordingly incorporated the mention in lines 625 – 628.

Comment 22: Fig: 7, whether the recovery of plants was observed for a longer duration. What were the results post 3days recovery?

Response: Thank you for this comment. We did indeed observed the recovery after 3 days (we decided to not include the data) and found that the wild-type tomato plants could not recover and showed no signs of growth, while the transgenic tomato plants could recover (side buds were exposed in the stem, and middle and bottom leaves were restored) only when the apex remained intact. 

Comment 23: Levels of SiDHN was found highest in OE8, but levels of most of the antioxidant enzymes including tolerance and survival under stress were observed in OE2 (Recovery). How would the author explain it? Similarly, in the case of expression of analysis of APX, SOD, POX (Fig. 10) shows an increase in their levels with duration. But it’s difficult to understand unless it is shown that H2O2/O2- levels also increased.

Response: Thank you very much for this question. In fact, the expression level of SiDHN was highest in OE-2 (was lowest in OE-8; as can be seen in Fig 3) and is correspondent with its tolerance and survival under stress (e.g. Fig 7). Regarding the expression of APX, SOD and POX, we invite the reviewer to compare the levels of APX, SOD and POX at 48 h in Fig 10 with the levels of H2O2 and O2- (measured at 48 h) in Fig 9, particularly for OE-2: we can see that the levels of APX, SOD and POX were highest, while the levels of H2O2 and O2- were lowest. We do hope that the explanation is clear.

In addition to the corrections above, we have also checked and corrected other errors found during the revision.

---

## [Decision Letter · Decision Letter 1]

11 Oct 2019

PONE-D-19-17695R1

Overexpression of Saussurea involucrata  Dehydrin Gene SiDHN  Promotes Cold and Drought Tolerance in Transgenic Tomato Plants

PLOS ONE

Dear %Jianbo Zhu%,

Thank you for submitting your manuscript to PLOS ONE. After careful consideration, we feel that it has merit but does not fully meet PLOS ONE’s publication criteria as it currently stands. Therefore, we invite you to submit a revised version of the manuscript that addresses the points raised during the review process.

ACADEMIC EDITOR: The manuscript needs to be revised based on the comments of reviewer 1. 

We would appreciate receiving your revised manuscript by %within one month from the date of this letter%. To enhance the reproducibility of your results, we recommend that if applicable you deposit your laboratory protocols in protocols.io, where a protocol can be assigned its own identifier (DOI) such that it can be cited independently in the future. For instructions see: http://journals.plos.org/plosone/s/submission-guidelines#loc-laboratory-protocols

We look forward to receiving your revised manuscript.

Kind regards,

T. R. Ganapathi, Ph.D.

Academic Editor

PLOS ONE

Additional Editor Comments (if provided):

Revise the manuscript based on the comments of reviewer 1.

Reviewers' comments:

Reviewer's Responses to Questions

**Comments to the Author**

1. If the authors have adequately addressed your comments raised in a previous round of review and you feel that this manuscript is now acceptable for publication, you may indicate that here to bypass the “Comments to the Author” section, enter your conflict of interest statement in the “Confidential to Editor” section, and submit your "Accept" recommendation.

Reviewer #1: (No Response)

Reviewer #2: All comments have been addressed

2. Is the manuscript technically sound, and do the data support the conclusions?

Reviewer #1: Yes

Reviewer #2: Yes

3. Has the statistical analysis been performed appropriately and rigorously? 

Reviewer #1: Yes

Reviewer #2: Yes

4. Have the authors made all data underlying the findings in their manuscript fully available?

Reviewer #1: Yes

Reviewer #2: Yes

5. Is the manuscript presented in an intelligible fashion and written in standard English?

Reviewer #1: Yes

Reviewer #2: Yes

6. Review Comments to the Author

Reviewer #1: Comments

Title of Manuscript: Overexpression of Saussurea involucrata Dehydrin Gene SiDHN Promotes Cold and Drought Tolerance in Transgenic Tomato Plants

The authors have made efforts to sufficiently revised manuscript based on the comments. However, still there are several shortcomings as mentioned below, which need to be addressed by the authors before acceptance of the manuscript.

1. Abstract: The term “tolerance” appears to be more appropriate instead of “resistance” for improved performances against abiotic stresses. Please check line no 54-56.

2. Materials and methods:

Line 111: Plantlets of S. involucrata were prepared by tissue culture. It is not clear why authors followed tissue culture to produce plant lets. Moreover, progeny obtained through indirect regeneration may not be true to type due to somaclonal variations. Whether the obtained plantlets were not hardened?

Line No 117: NAA (N-acetyl-aspartate); Please check full form of NAA.

Lines 123-124: Tomato plant variety ‘Yaxin 87-5’ used as transgenic plants were grown from 124 seeds provided by Yaxin Seed Co. Ltd. (Shihezhi City, Xinjiang, China). Please rewrite the sentence, as meaning is not clear. Whether, genetic transformation of tomato was carried out by the authors or transgenic seeds provided by Yaxin Seed Co. Ltd were used to grow the transgenic plants for the present study?

Line 126-127: After that, they were sown in plastic pots ....Do you mean the plantlets were transplanted in plastic pots....

Line 146-148: Top-second and -third leaves of the plants were harvested at 4°C and 0 °C after the 48 h treatment and at 0, 16 and 24 d after the 148 drought treatment. Meaning of the sentence is not clear.

Line 217-218: ....and the obtained plasmid was regenerated in Escherichia coli DH5 alpha. Do you mean “the obtained plasmid was transformed in Escherichia coli strain DH5α ?”

Line No. 221-223. To produce transgenic potato plants, the wild-type tomato plants were transformed ….. What do you mean? Transgenic potato plants were developed after transforming tomato?

Lines 258 – 260: Measurement of chlorophyll pigment content:

....the supernatant was subjected to absorbance measurements at 470 nm, 645 nm, and 663 nm to determine the contents of chlorophyll a (Chl a), chlorophyll b (Chl b), and carotenoids (Car), respectively. Do you mean, the estimations of chl a, chl b and car are based on absorbance at single wavelength?

Line No 318: .....for 10 min, 200 mL of the supernatant was mixed with 1.8.... Is it 200mL or 200µl?

3. Results: Author should elaborate the results on effect of cold and drought treatments on physio-biochemical traits in wild type and transgenic plants. Further, relation between the transcript level and activity of the coded anti-oxidant enzymes need to be stated in detail.

Fig 5,6: X Axis title: Instead of “Different strains”, “Different lines” is appropriate.

4. Discussion:

Line No 651-652:..... in protecting the transgenic potato plants against cold and drought stress. Potato or tomato?

Recommendation: Major revision

Reviewer #2: (No Response)

7. PLOS authors have the option to publish the peer review history of their article (what does this mean?). If published, this will include your full peer review and any attached files.

Reviewer #1: No

Reviewer #2: No

---

## [Author Response · Author response to Decision Letter 1]

24 Oct 2019

October 24, 2019

T. R. Ganapathi, Ph.D.

Academic Editor 

PLOS ONE

Dear editor:

We highly appreciate you and the reviewers for reviewing our manuscript and giving us the opportunity to submit a revised draft of our manuscript titled “Overexpression of Saussurea involucrata Dehydrin Gene SiDHN Promotes Cold and Drought Tolerance in Transgenic Tomato Plants” to PLOS ONE. We also appreciate your and the reviewers’ valuable and insightful feedback on our manuscript. 

We have carefully considered the reviewer’s comments and respond to all of the suggested revisions in more detail below. The reviewers’ comments are italics and our responses are in plain text; line and page numbers refer to the revised manuscript. 

Comments from Reviewer 1

Comment 1: Abstract: The term “tolerance” appears to be more appropriate instead of “resistance” for improved performances against abiotic stresses. Please check line no 54-56.

Response: Thank you for this comment. We agree to the reviewer and have changed the term accordingly. Please see line 56.

Comment 2: Materials and methods: 

Comment: Line 111: Plantlets of S. involucrata were prepared by tissue culture. It is not clear why authors followed tissue culture to produce plantlets. Moreover, progeny obtained through indirect regeneration may not be true to type due to somaclonal variations. Whether the obtained plantlets were not hardened?

Response: Thank you for this comment. We followed the tissue culture to produce plantlets due to the following reasons:

1.Due to its special growth conditions, artificial propagation of Saussurea involucrata is very difficult. Because plant tissue culture technology is not limited by time, place, climate and other environmental conditions, the culture conditions can be easily controlled and can achieve long-term preservation. This not only allows researchers to produce the plants at a large scale at any time, but also provides an important way for mass propagation of medicinal plant S. involucrata.

2.The sequence of SiDHN was obtained by sequencing of the full-length cDNA library of Saussurea involucrata. And the materials used to build the library were obtained by tissue culture seedling that has been preserved in our laboratory for a long period of time. More importantly, the cloning of SiDHN used in this study was from the same tissue culture and materials.

3.We compared the cDNA sequence of SiDHN gene cloned in this paper with the full-length cDNA sequence of Saussurea involucrata from the database, and found that they were 100% identical. This should be an indication that our method is reliable.

4.This approach has been widely reported. Please see below for example publications that have conducted similar experiments and have used similar methods.

Xinyong Guo, Li Zhang, Gaoquan Dong, Zhihua Xu, Guiming Li, Ning Liu, Aiying Wang, Jianbo Zhu. A novel cold-regulated protein isolated from Saussurea involucrata confers cold and drought tolerance in transgenic tobacco (Nicotiana tabacum). Plant Science, 2019, 289: 110246. 

Linhua Zhang, Li Sun, Li Zhang, Hongling Qiu, Chao Liu, Aiying Wang, Jianbo Zhu. A Cu/Zn superoxide dismutase gene from Saussurea involucrata Kar. & Kir., SiCSD, enhances drought, cold, and oxidative stress in transgenic tobacco, Canadian Journal of Plant Science. 2017, 97(5): 816-826.

Liu HL, Shen HT, Chen C, Zhou XR, Liu H, and Zhu JB. Identification of a putative stearoyl acyl-carrier-protein desaturase gene from Saussurea involucrata. Biologia Plantarum. 2015, 59 (2): 316-324. 

Comment: Line No 117: NAA (N-acetyl-aspartate); Please check full form of NAA.

Response: Thank you for pointing this out. We have changed it to the correct name, which is ‘1-Naphthaleneacetic acid’. Please see line 117.

Comment: Lines 123-124: Tomato plant variety ‘Yaxin 87-5’ used as transgenic plants were grown from 124 seeds provided by Yaxin Seed Co. Ltd. (Shihezhi City, Xinjiang, China). Please rewrite the sentence, as meaning is not clear. Whether, genetic transformation of tomato was carried out by the authors or transgenic seeds provided by Yaxin Seed Co. Ltd were used to grow the transgenic plants for the present study?

Response: Thank you for this comment. Yaxin Seed Co. Ltd. provided the wild-type tomato plant seeds (variety "Yaxin 87-5"), not the transgenic tomato plants/seeds. We then grew tomato plants from the seeds (denoted as wild-type tomato plants), and the plants obtained were used to generate the transgenic tomato plants (denoted as transgenic tomato plants). 

Accordingly, we have changed the sentence to the following: ‘Seeds of tomato plant variety ‘Yaxin 87-5’ (wild-type) were provided by Yaxin Seed Co. Ltd. (Shihezhi City, Xinjiang, China). The wild-type tomato plants were grown from the seeds in our laboratory and were used to produce transgenic tomato plants.’ Please refer to lines 123-126.

We do hope that it is now clearer.

Comment: Line 126-127: After that, they were sown in plastic pots ....Do you mean the plantlets were transplanted in plastic pots....

Response: Thank you for the comment. That is correct. Accordingly, we have changed the sentence to ‘After that, the plantlets were transplanted in plastic pots.’ line 128.

Comment: Line 146-148: Top-second and -third leaves of the plants were harvested at 4°C and 0 °C after the 48 h treatment and at 0, 16 and 24 d after the 148 drought treatment. Meaning of the sentence is not clear.

Response: Thank you for this comment. Please note that there should be no ‘148’ in the manuscript; we are not sure if this number shows in the reviewer’s file.

We collected the second and the third leaves from the top of the plants. The leaves of the cold-treated plants (at both temperatures) were collected at 48 h, which is the end of the treatment, and the leaves of the drought-treated plants were collected at day 0, 16, and 24. 

We have changed the sentence to: ‘Tomato leaves (the second and third leaves from the top) were collected after 48 h of the cold treatments (4oC and 0oC), or after 0, 16 and 24 d of the drought treatment.’ We do hope that it is now clearer. Please see lines 148-150.

Comment: Line 217-218: ....and the obtained plasmid was regenerated in Escherichia coli DH5 alpha. Do you mean “the obtained plasmid was transformed in Escherichia coli strain DH5α ?”

Response: Thank you for this comment. Yes, it is in fact more accurate to write ‘… and the obtained plasmid was transformed into Escherichia coli strain DH5α’. We have changed the sentence accordingly (lines 218-219).

Comment: Line No. 221-223. To produce transgenic potato plants, the wild-type tomato plants were transformed ….. What do you mean? Transgenic potato plants were developed after transforming tomato?

Response: We apologize for this error and thank you for pointing it out. All should be ‘tomato’, and we have changed all accordingly. We have also ensured that the word ‘potato’ no longer exists in our manuscript. Thank you very much again. 

Comment: Lines 258-260: Measurement of chlorophyll pigment content: ....the supernatant was subjected to absorbance measurements at 470 nm, 645 nm, and 663 nm to determine the contents of chlorophyll a (Chl a), chlorophyll b (Chl b), and carotenoids (Car), respectively. Do you mean, the estimations of chl a, chl b and car are based on absorbance at single wavelength?

Response: Thank you for this question. We determined the contents of Chl a, Chl b, and Car using the wavelengths described in the sentence. And the contents were calculated by the following equations: 

Chl a = 12.21*A663 – 2.81*A645

Chl b = 20.13*A645 – 5.03*A663

Car = (1000*A470 – 3.27*Chl a – 104*Chl b)/229.

Accordingly, we have changed the content to ‘After centrifugation, the supernatant was subjected to absorbance measurements at 470 nm, 645 nm, and 663 nm, and the contents of chlorophyll a (Chl a), chlorophyll b (Chl b), and carotenoids (Car) were calculated by the following equations: Chl a = 12.21*A663 – 2.81*A645; Chl b = 20.13*A645 – 5.03*A663; and Car = (1000*A470 – 3.27*Chl a – 104*Chl b)/229.’ Please refer to lines 262-264.

Comment: Line No 318: .....for 10 min, 200 mL of the supernatant was mixed with 1.8.... Is it 200mL or 200µl?

Response: We appreciate the comment and apologize for the error. This should in fact be 2 mL (not 200 mL). We have corrected the sentence accordingly (line 324). 

Comment 3: Results: Author should elaborate the results on effect of cold and drought treatments on physio-biochemical traits in wild type and transgenic plants. Further, relation between the transcript level and activity of the coded anti-oxidant enzymes need to be stated in detail. Fig 5,6: X Axis title: Instead of “Different strains”, “Different lines” is appropriate.

Response: Thank you for this comment. The study showed that overexpression of SiDHN alleviated the accumulation of ROS by upregulating the activity of ROS scavenging enzymes SOD, POD, CAT and POX under low temperature and drought stress conditions (Figs. 10 and 11). To study why transgenic plants can maintain relatively high antioxidant enzyme activity under low temperature and drought stress, we measured the expressions of related genes encoding SOD, POD, CAT and APX. RT-PCR results showed that the expressions of SlSOD, SlPOD, SlCAT and SlAPX were significantly up-regulated in SiDHN transgenic plants OE-2, OE-5 and OE-8 (Figs. 12 and 13). These results indicate that overexpression of SiDHN may be able to improve the expression of antioxidant related genes, thus increasing the activity of the corresponding antioxidant enzymes, enhancing ROS clearance, and in turn reducing cell membrane damage caused by low temperature- and drought-induced oxidative stress.

We have incorporated similar details as those outlined above into the manuscript. We invite the reviewer to check the revised paragraph in pages 17 and 18, and we hope that it is now clearer and sufficient. 

Regarding Figs. 5 and 6, we have accordingly changed the X-axis title (from ‘Different strains’ to ‘Different lines’). 

Comment 4: Discussion: Line No 651-652:..... in protecting the transgenic potato plants against cold and drought stress. Potato or tomato?

Response: We sincerely apologize for this error and thank you very much for pointing it out. All should be ‘tomato’, and we have changed all accordingly. We have also ensured that the word ‘potato’ no longer exists in our manuscript. 

In addition to the corrections above, we have also checked and corrected other errors found during the revision. 

We do hope that you find our responses and corrections satisfactory. We look forward to hearing from you regarding our submission and are pleased to respond to any further questions and comments you may have.

Sincerely,

Jianbo Zhu

Key Laboratory of Agricultural Biotechnology

College of Life Science, Shihezi University, 

Shihezi, Xinjiang, China 

Phone No: 18935701090

Fax No: 18935701090

E-mail Address: jianboz9@sina.com

---

## [Decision Letter · Decision Letter 2]

30 Oct 2019

Overexpression of Saussurea involucrata  Dehydrin Gene SiDHN  Promotes Cold and Drought Tolerance in Transgenic Tomato Plants

PONE-D-19-17695R2

Dear Dr. Jianbo Zhu,

We are pleased to inform you that your manuscript has been judged scientifically suitable for publication and will be formally accepted for publication once it complies with all outstanding technical requirements.

With kind regards,

T. R. Ganapathi, Ph.D.

Academic Editor

PLOS ONE

Additional Editor Comments (optional):

The manuscript is accepted for publication.

Reviewers' comments:

Reviewer's Responses to Questions

**Comments to the Author**

1. If the authors have adequately addressed your comments raised in a previous round of review and you feel that this manuscript is now acceptable for publication, you may indicate that here to bypass the “Comments to the Author” section, enter your conflict of interest statement in the “Confidential to Editor” section, and submit your "Accept" recommendation.

Reviewer #1: All comments have been addressed

2. Is the manuscript technically sound, and do the data support the conclusions?

Reviewer #1: Yes

3. Has the statistical analysis been performed appropriately and rigorously? 

Reviewer #1: Yes

4. Have the authors made all data underlying the findings in their manuscript fully available?

Reviewer #1: Yes

5. Is the manuscript presented in an intelligible fashion and written in standard English?

Reviewer #1: Yes

6. Review Comments to the Author

Reviewer #1: The manuscript has been substantially revised by the authors to address all the comments.

Hence the revised manuscript is recommended for publication.

7. PLOS authors have the option to publish the peer review history of their article (what does this mean?). If published, this will include your full peer review and any attached files.

Reviewer #1: No

---

## [Editor Report · Acceptance letter]

8 Nov 2019

PONE-D-19-17695R2 

Overexpression of *Saussurea involucrata* Dehydrin Gene *SiDHN* Promotes Cold and Drought Tolerance in Transgenic Tomato Plants

Dear Dr. Zhu:

I am pleased to inform you that your manuscript has been deemed suitable for publication in PLOS ONE. Congratulations! Your manuscript is now with our production department. 

With kind regards,

on behalf of

Dr. T. R. Ganapathi 

Academic Editor

PLOS ONE